# A causal role of the NMDA receptor in recurrent processing during perceptual integration

Samuel Noorman[1,2]*, Timo Stein[1,2], Jasper Zantvoord[3,4], Johannes Fahrenfort[1,2,5†], Simon van Gaal[1,2†]

[1]Department of Psychology, University of Amsterdam, Amsterdam, Netherlands; [2]Amsterdam Brain and Cognition, University of Amsterdam, Amsterdam, Netherlands; [3]Department of Psychiatry, Amsterdam UMC, University of Amsterdam, Amsterdam, Netherlands; [4]Amsterdam Neuroscience, Amsterdam, Netherlands; [5]Department of Applied and Experimental Psychology, Vrije Universiteit Amsterdam, Amsterdam, Netherlands

*For correspondence: sgnoorman@gmail.com

†These authors contributed equally to this work

## eLife Assessment

This study presents a new, **fundamental** finding to the field interested in recurrent processing and its neuromodulatory underpinnings, finding unexpectedly that memantine (blocking NMDA-receptors) enhances the decoding of features thought to rely on NMDA-receptors. This interesting, **compelling** result identifies new directions for researchers studying consciousness, sensory processing, attention, and neurotransmitters.

**Abstract** Perceptual inference requires the integration of visual features through recurrent processing, the dynamic exchange of information between higher- and lower-level cortical regions. While animal research has demonstrated a crucial role of NMDA receptors in recurrent processing, establishing a causal link between NMDA receptors and recurrent processing in humans has remained challenging. Here, we report two pharmacological studies with randomized, double-blind, crossover designs in which we administered the NMDA antagonist memantine, while collecting human electroencephalography (EEG). We trained and tested EEG classifiers to reflect the processing of specific stimulus features with increasing levels of complexity, namely differences in stimulus contrast, collinearity between local line elements, and illusory surfaces of a Kanizsa triangle. In two experiments involving different participants and visual tasks, we found that memantine selectively improved decoding of the Kanizsa illusion, known to depend on recurrent processing, while leaving decoding of contrast and collinearity largely unaffected. Interestingly, the results from an attentional blink (experiment 1) and task-relevance manipulation (experiment 2) showed that memantine was only effective when the stimulus was attended and consciously accessed. These findings suggest that NMDA inhibition through memantine enhances recurrent processing, especially for attended objects, and thereby provide a crucial step toward bridging animal and human research, shedding light on the neural mechanisms underpinning perceptual inference and conscious perception.

## Introduction

Different theories of consciousness converge on a central principle: consciousness entails integration of information through recurrent neural processes that exceeds the capabilities of mere feedforward

processing (*Mashour et al., 2020*; *Seth and Bayne, 2022*). Recurrent neural mechanisms encompass feedback connections between higher- and lower-order cortical areas as well as lateral connections within sensory areas. Studies in mice and monkeys have revealed a causal role of feedback from higher cortical areas to sensory areas for tactile perception (*Manita et al., 2015*; *Sachidhanandam et al., 2013*; *Takahashi et al., 2016*), detection of odors (*Allen et al., 2017*), and visual stimulus attributes (*Ciceri et al., 2023*; *Lamme et al., 1998*; *van Vugt et al., 2018*; *Zatka-Haas et al., 2021*). In macaques, general anesthetics targeted at reducing the overall level of consciousness of the animal reduce global cortical recurrent interactions (*Mashour and Hudetz, 2018*) and block feedback connections to V1, while sparing feedforward processing (*Lamme et al., 1998*). Although in humans indirect evidence for a role of recurrent processing in consciousness and perceptual integration has been provided, for example by focusing on neural markers of figure-ground segregation or backward masking (*Fahrenfort et al., 2007*; *Fahrenfort et al., 2008*; *Fahrenfort et al., 2012*; *Fahrenfort et al., 2017*; *Kok et al., 2016*; *Kok and de Lange, 2014*; *Noorman et al., 2023*; *Scholte et al., 2008*; *Wyatte et al., 2014*), the underlying biochemistry has remained so far elusive. In this study, we aimed to provide direct evidence for a causal role of NMDA receptors in recurrent processing (through lateral and feedback connections) and their role in perceptual inference and conscious access, factoring in potential modulating influences of attention (*Flohr et al., 1998*; *Lamme, 2010*; *Sachidhanandam et al., 2013*; *Self et al., 2012*; *Thiele, 2012*; *Thiele and Bellgrove, 2018*; *van Kerkoerle et al., 2014*; *Wang, 2008*).

NMDA has a slower time constant than most other receptors (decay time constant ~100 ms, voltage-gated) and is known to be densely populated in recurrent connections in the cortex (*Wang, 2008*). In macaque monkeys, blocking NMDA receptors selectively disrupted late onset modulations of neural activity in V1 (i.e., recurrent processing) during figure-ground detection, whereas blocking AMPA receptors did not, but instead reduced the strength of the first feedforward sweep (*Self et al., 2012*; *van Kerkoerle et al., 2014*). This suggests that feedback connections target neuronal synapses that are rich in NMDA receptors (among other receptors), whereas feedforward connections target synapses especially rich in AMPA receptors (*Thiele, 2012*; *Thiele and Bellgrove, 2018*). In mice, blocking NMDA receptors strongly reduced hit rates for the detection of whisker movements (*Sachidhanandam et al., 2013*). In humans, few studies have investigated the role of NMDA receptors in recurrent processing and (conscious) perception. Blocking NMDA receptors via administration of ketamine resulted in impaired feature integration in a texture discrimination task (*Meuwese et al., 2013*) and a distortion of the neural representations of objects in early visual cortex (*van Loon et al., 2016*). However, as these human studies did not clearly distinguish between markers of feedforward and recurrent processing, they could not determine whether these effects were specific to recurrent processing or reflected other non-specific effects. Ketamine has rather broad adverse side effects, such as sedation and dissociation, which could potentially explain the observed behavioral deficits and neural distortions. Also, ketamine is known for its complex pharmacological dynamics, having effects on several neurotransmitter systems, including catecholaminergic, cholinergic, nicotinic, muscarinic, and GABAergic receptors (*Kocsis et al., 2013*; *Mathew et al., 2012*; *Müller et al., 2011*).

In this study, we used a novel Kanizsa stimulus whose different visual features were designed to target distinct processing stages with increasing levels of complexity, namely local contrast differences, differences in collinearity of local line elements, and the presence/absence of an illusory triangle (*Figure 1A*; see also *Noorman et al., 2023*). The processing of collinearity of line elements predominantly relies on lateral connections (*Bosking et al., 1997*; *Gilbert and Wiesel, 1979*; *Li, 1998*; *Liang et al., 2017*; *Schmidt et al., 1997*; *Stettler et al., 2002*), while processing of the illusory surface of the Kanizsa illusion relies on feedback connections (*Halgren et al., 2003*; *Kok et al., 2016*; *Kok and de Lange, 2014*; *Lee and Nguyen, 2001*; *Pak et al., 2020*; *Wokke et al., 2013*). Therefore, multivariate EEG decoding of the presence/absence of the illusionary Kanizsa triangle served as a marker of feedback processing (*Fahrenfort et al., 2017*; *Noorman et al., 2023*). The ancillary features of the stimulus, the two-legged white circles, could be aligned to form a non-illusionary triangle, enabling us to use collinearity decoding as a marker of lateral processing. By rotating the Kanizsa by 180°, thus either having one black Pacman on top of the screen or two, we created differences in contrast between the upper and lower visual field. Because local contrast differences are processed early in the visual system, this allowed us to employ local contrast decoding as a marker of feedforward processing. In recent work, we have validated that the decoding profiles of these features of different complexities

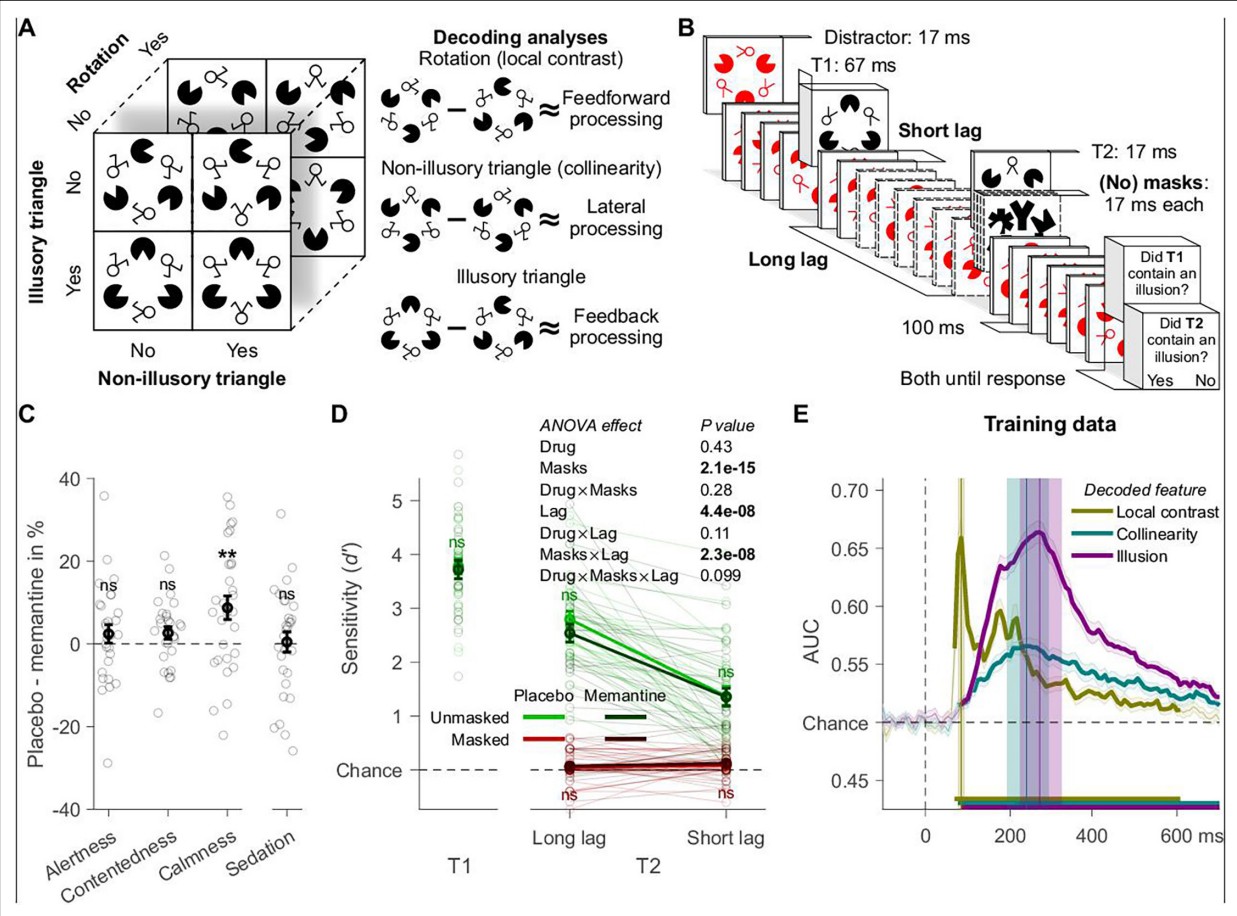

**Figure 1.** Experiment 1: design, behavior, and training data. (**A**) Target stimulus set and schematic of the markers for the different types of processing (showing example stimuli). (**B**) Trial design. (**C**) Effects of memantine on subjective state assessed with the visual analog scale (*n* = 29). The categories alertness, contentedness, and calmness are based on *Bond and Lader, 1974*, sedation on *Danion et al., 1989*. For details, see Methods. (**D**) Participants' ability to detect the Kanizsa illusion in the second target stimulus (T2). (**E**) Tenfold cross-validation decoding scheme applied to an independent training dataset. Mean decoding performance, area under the receiver operating characteristic curve (AUC), over time ± standard error of the mean (SEM) is shown. Thick lines differ from chance: p < 0.05, cluster-based permutation test. Local contrast, collinearity, and illusion decoding accuracy peaked at 86, 242, and 273 ms after stimulus onset, and these peaks were encompassed by a 78–94, 192–292, and 223–323 ms time window, respectively. Error bars are mean ± SEM. Individual data points are plotted using low contrast. Ns is not significant (p > 0.05). **p = 0.005.

at different points in time, in combination with the associated topography, can indeed serve as EEG markers of feedforward, lateral, and feedback processes (*Fahrenfort et al., 2017*; *Noorman et al., 2023*).

We tested how these neural markers were affected by memantine, a low-affinity, voltage-dependent, uncompetitive NMDA antagonist. Memantine is well tolerated and has minimal side effects compared to ketamine.

We performed two separate experiments. In experiment 1, we perturbed conscious access through masking and inattention using a modified attentional-blink (AB) paradigm (*Figure 1B*). In experiment 2, we manipulated whether the illusion, collinearity, or local contrast was task-relevant or not. We hypothesized that disrupting the reentrant glutamate signal via blocking NMDA receptors by memantine would impair illusion and possibly collinearity decoding, as putative markers of feedback and lateral processing, but would spare the decoding of local contrast differences, our marker of feedforward processing. To foreshadow our results, memantine indeed specifically affected illusion decoding, but enhancing rather than impairing it. In the Discussion, we offer explanations for this surprising finding, including the effect of memantine on extrasynaptic NMDA receptors in GABAergic cells, which may have resulted in boosted visual activity.

# Results

## Memantine caused minimal side effects

The study had a randomized, double-blind, crossover design in which a placebo was administered during one session and memantine (20 mg) during another. Drug order was counterbalanced across the 29 healthy participants. Potential side effects of memantine on different aspects of the participant's subjective state were assessed with the visual analog scale (VAS) (*Bond and Lader, 1974*; *Danion et al., 1989*). The only significant effect was a moderate decrease in the category 'calmness' for memantine compared to placebo ($t_{28}$ = −23.07, p = 0.005, $BF_{10}$ = 8.71; *Figure 1C*, for details see Methods). Note that in experiment 2 (Figure 4), none of these subjective measures differed between memantine and placebo, indicating that overall subjective side effects of memantine were minimal.

## Memantine selectively improves decoding of the Kanizsa illusion

We recorded EEG signals from participants who detected illusory surfaces in two black targets (T1 and T2) among red distractors in a rapid serial visual presentation (RSVP) task (*Figure 1B*). Visibility of T2 was manipulated in two ways, resulting in a 2 × 2 factorial task design. First, T2 was either masked or unmasked. Second, shortening the interval (lag) between the two targets from 900 to 200/300 ms induced an AB, which reduced T2 visibility as well. At the end of a trial, participants indicated for each target whether they had perceived an illusory surface or not. For all main analyses of behavior and EEG decoding, we conducted repeated-measures ANOVAs with the factors drug (memantine vs. placebo), T1–T2 lag (short, inducing an AB, vs. long, no AB), and masks (masked vs. unmasked). The complete results of these ANOVAs are presented in the corresponding results figures. In the text, we only highlight the key results and report selectively additional statistics such as post hoc tests.

As expected, both masking and the AB impaired the participant's ability to detect the Kanizsa illusion in the second target (T2). Our masking procedure was highly effective, leading to poor T2 performance on masked trials (red lines in *Figure 1D*). The AB was also effective, leading to a sharp drop in T2 performance on short lag versus long lag trials (green lines in *Figure 1D*). Memantine had, however, no effect on perceptual sensitivity (*Figure 1D*).

To establish our markers for the different neural processes of interest, the processing of different visual features (local contrast, collinearity, and the Kanizsa illusion) was decoded (*Figure 1A*). For

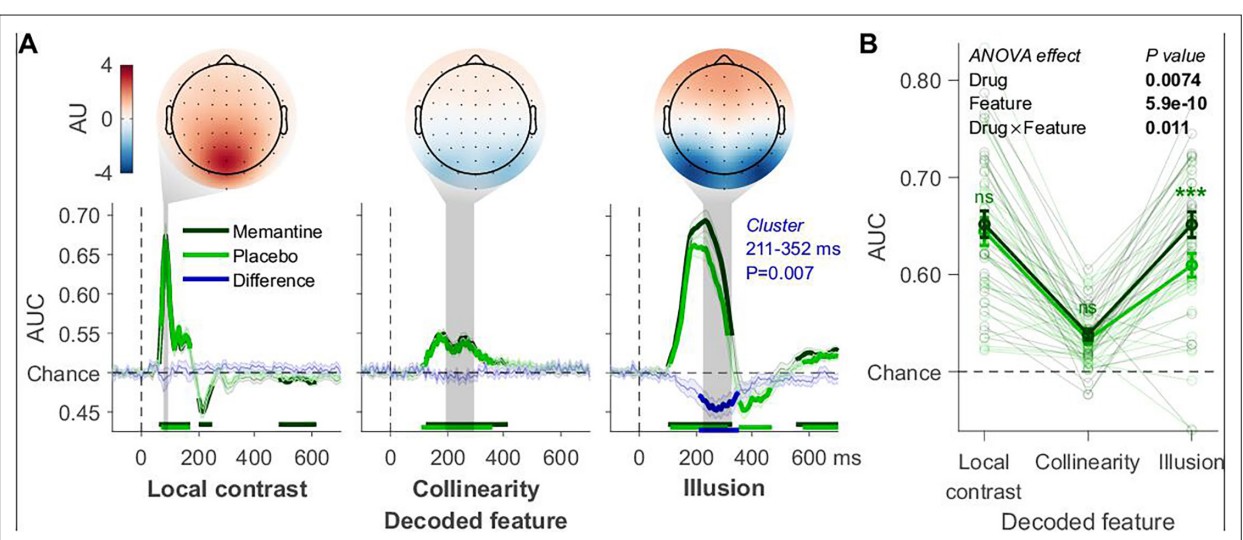

**Figure 2.** Decoding applied to the T1 data. (**A**) Local contrast, collinearity, and Kanizsa illusion decoding. These three visual features' time windows: 78–94, 192–292, and 223–323 ms, respectively. Covariance/class separability maps with arbitrary units (AU) reflecting underlying neural sources are shown. Below these maps: mean decoding performance, area under the receiver operating characteristic curve (AUC), over time ± standard error of the mean (SEM; n = 29). Thick lines differ from chance: p < 0.05, cluster-based permutation test. (**B**) Mean AUC for every time window. Error bars are mean ± SEM. Individual data points are plotted using low contrast. Ns is not significant (p > 0.05). \*\*\*p < 0.001.

The online version of this article includes the following figure supplement(s) for figure 2:

**Figure supplement 1.** Diagonal decoding applied to the T1 data.

these decoding analyses, classifiers were trained using an independent training set obtained from a separate experimental task where the same stimuli were presented, but outside the context of the RSVP task (see Methods for details). No drugs were administered during this separate 'localizer' task. In brief, during the localizer, participants had to match the target stimulus to different example stimuli (templates). For successful performance in this template-matching task, all three features of the stimulus had to be attended, and all features were thus task-relevant. We also used the independent training set from this template-matching task to determine the time windows for local contrast, collinearity, and illusion decoding. First, we decoded each visual feature from the localizer using a tenfold cross-validation scheme by training linear discriminant classifiers to categorize the feature as either present or absent based on the raw EEG activity across all electrodes. After testing the classifiers, each decoded visual feature showed a clear peak in classifier performance (area under the receiver operating characteristic curve [AUC]) (*Hand and Till, 2001*) over time: local contrast at 86 ms after stimulus onset, collinearity at 242 ms, and the illusion at 273 ms. In the following, we refer to 'time window-based analyses' when we focus on the averages of the time windows encompassing these peaks (for the exact time windows used, see *Figure 1E*). The timing and mostly occipital nature (see the covariance/class separability maps of *Figure 2A*; *Haufe et al., 2014*) of these windows were consistent with the neural processes they are thought to reflect and in line with our previous EEG decoding studies (*Fahrenfort et al., 2017*; *Noorman et al., 2023*). Because of this topology, subsequent decoding analyses were restricted to occipitoparietal electrodes. To increase power, we averaged each time window in the localizer task's raw EEG data. Classifiers were trained on these averages and tested on all time samples of each condition of the main RSVP task's data (see Methods) (*King and Dehaene, 2014*). Note that this approach therefore reflects 'off-diagonal decoding' in the generalization across time approach. We show all 'on-diagonal' analyses in *Figure 2—figure supplement 1*, *Figure 3—figure supplement 1*, and *Figure 5—figure supplement 1*, which revealed very similar results.

We first examined the effect of memantine on the decoding of the different features of the first target (T1) from the RSVP task. For each time sample, we tested whether the memantine conditions differed in classifier performance from the placebo conditions. Cluster-based permutation testing was used to correct for multiple comparisons (*Maris and Oostenveld, 2007*). Crucially, only illusion decoding showed a significant cluster ranging from 211 to 352 ms after stimulus onset (*Figure 2A*, blue lines). A similar result was obtained from the time window-based analysis (*Figure 2B*, ANOVA results in the figure). Memantine significantly improved overall T1 decoding ($F_{1,28} = 8.33$, p = 0.007) and drug interacted with visual feature ($F_{1,28} = 5.16$, p = 0.011). Post hoc *t*-tests confirmed that memantine selectively affected illusion decoding accuracy ($t_{28} = -4.14$, p < 0.001, $BF_{10} = 101.92$), thus leaving local contrast ($t_{28} = -0.60$, p = 0.560, $BF_{10} = 0.23$) and collinearity decoding unaffected ($t_{28} = -1.04$, p = 0.306, $BF_{10} = 0.32$). As memantine is an NMDA blocker, we had initially hypothesized that it would impair recurrent processing, and therefore decrease illusion decoding accuracy. Surprisingly, however, although memantine specifically affected illusion decoding, it *increased* instead of decreased illusion decoding. Recent studies using memantine have, however, mostly observed increases in task performance and/or neural markers of sensory processing (*Korostenskaja et al., 2007*; *Light et al., 2017*; *Swerdlow et al., 2016*; *Swerdlow et al., 2020*), and are therefore mostly in line with our observations here. To foreshadow our further results, this effect was robust as we obtained the same *increase* in Kanizsa decoding for T2s in this study (*Figure 3*), and the effect was replicated in study 2 (Figure 5). We address this issue further in the discussion.

## The role of consciousness and attention in memantine's effect on illusion decoding

To determine how memantine interacted with our manipulations of consciousness (masking) and attention (T1–T2 lag, i.e., AB), we examined the decoding of the different features of the second target (T2) from the RSVP task (*Figure 3A, C, E*). First, we tested whether masking and the AB affected local contrast, collinearity, and illusion decoding in the way we expected. Indeed, local contrast decoding was unaffected by the AB (*Figure 3B*), and although it was impaired by masking, the drop in decoding accuracy was numerically small (0.012 AUC on average) and considerably smaller than the effect of masking on collinearity (0.026 AUC) and illusion (0.123 AUC) decoding. These results are in line with theoretical proposals and empirical findings that suggest limited effects of masking and attentional

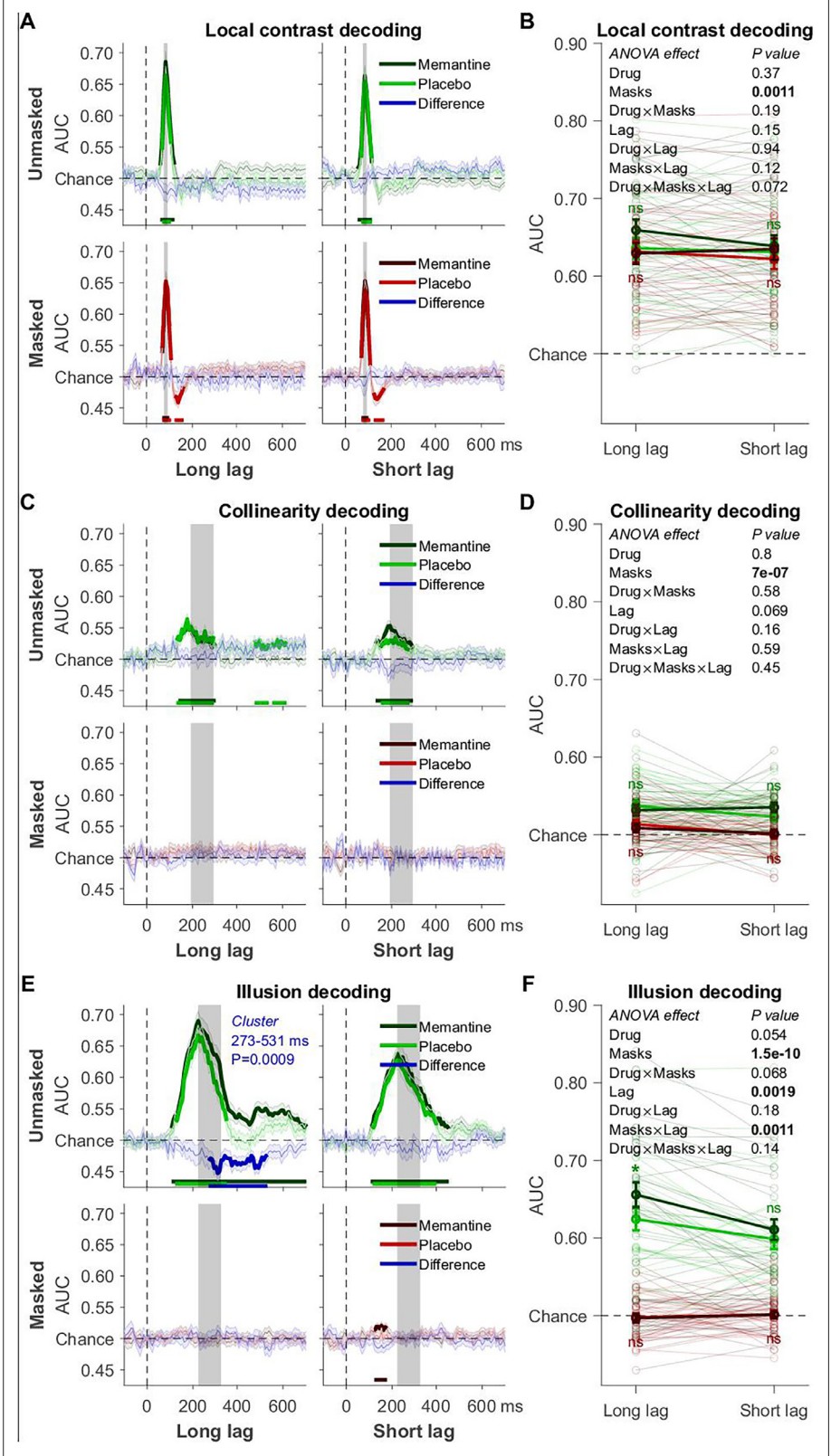

**Figure 3.** Decoding of the T2 data. (**A**) Local contrast decoding: mean decoding performance, area under the receiver operating characteristic curve (AUC), over time ± standard error of the mean (SEM; $n = 29$). Thick lines differ from chance: $p < 0.05$, cluster-based permutation test. (**B**) Mean AUC for local contrast (rotation) decoding's time window: 78–94 ms. Error bars are mean ± SEM. Individual data points are plotted using low contrast. Ns is

*Figure 3 continued on next page*

*Figure 3 continued*

not significant (p > 0.05). (**C**) Collinearity decoding: mean AUC over time. (**D**) Mean AUC for collinearity decoding's time window: 192–292 ms. (**E**) Kanizsa illusion decoding: mean AUC over time. (**F**) Mean AUC for Kanizsa illusion's time window: 223–323 ms. *p = 0.010.

The online version of this article includes the following figure supplement(s) for figure 3:

**Figure supplement 1.** Diagonal decoding of the T2 data.

manipulations on feedforward processes (***Dehaene et al., 2006***; ***Fahrenfort et al., 2007***; ***Fahrenfort et al., 2017***; ***Lamme, 2010***; ***Lamme et al., 2002***). As would be expected for markers of lateral and feedback processing, both collinearity and illusion decoding were abolished by masking (***Figure 3C, F***). The AB exclusively impaired illusion decoding, but decoding accuracy for the illusion remained well above chance during the AB (***Figure 3E, F***).

Similar to T1, also for T2 decoding, memantine increased illusion decoding, but only when T2 was presented outside the AB and when it was unmasked (***Figure 3E***, top left, note that masking fully abolished illusion decoding). Also, our time window-based analyses showed a significant effect of memantine only when the illusion was both unmasked and presented outside the AB ($t_{28} = -2.76$, $p = 0.010$, $BF_{10} = 4.53$; ***Figure 3F***). Note, however, that although these post hoc tests of the effect of memantine on illusion decoding were significant, for our time window-based analyses we did not obtain a statistically significant interaction between the AB and memantine. While the interaction between masking and memantine only approached significance ($p = 0.068$), the absence of an effect of memantine in the masked condition could reflect a floor effect, given that illusion decoding in the masked condition was not significantly better than chance. Thus, although these memantine effects were slightly less robust than for T1, probably due to reduced trial counts, these results point to (but do not conclusively demonstrate) a selective effect of memantine on illusion-related feedback processing that depends on the availability of attention. In addition to the lack of the interaction effect, another potential concern with this interpretation is that in the RSVP task, the illusion was always task-relevant and hence attended, whereas local contrast and collinearity were never task-relevant. It is therefore an open question whether local contrast and collinearity decoding were unaffected by memantine, because these visual features were unattended (not task-relevant) and/or because memantine exclusively targets feedback processes. To alleviate this potential concern and to replicate the surprising result of increased illusion decoding under memantine, we carried out a second experiment. A new group of participants was recruited and we systematically varied task relevance of local contrast, collinearity, and the illusion in this task. This allowed us to determine whether memantine indeed specifically affected feedback processing.

## Experiment 2 replicates memantine's selective effect on illusion decoding

In experiment 2, the pharmacological design was the same as before, and VAS measurements now revealed no side effects from memantine (***Figure 4B***). Twenty-seven new participants performed a discrimination task in which each visual feature, in separate blocks, was task-relevant (***Figure 4A***). Approximately every 900–1100 ms an individual target stimulus (no RSVP) was presented for 83 ms when it was unmasked, or for 33 ms when it was masked. The timing of the masked condition was increased compared to experiment 1 to circumvent floor or ceiling effects. Performance on the Kanizsa illusion was staircased to 75% correct in a practice session by varying mask contrast, and these mask settings were used for all features (see Methods). For each target, participants indicated whether the current task-relevant feature (e.g., the Kanizsa illusion) was present or absent. In experiment 2, each visual feature was task-relevant and thus attended in some blocks, while it was task-irrelevant and thus less attended in other blocks. This allowed us to directly examine whether memantine's effects were specific to feedback processes or dependent on the task relevance of the stimulus features. For all main analyses of behavior and EEG decoding, we conducted repeated-measures ANOVAs with the factors drug (memantine vs. placebo), masking (masked vs. unmasked), and task-relevant feature (local contrast, collinearity, and illusion). The complete results of these ANOVAs are presented in the corresponding results figures. We found that masking impaired perceptual sensitivity, most prominently for the detection of the Kanizsa illusion, and less so for collinearity and rotation. Note that

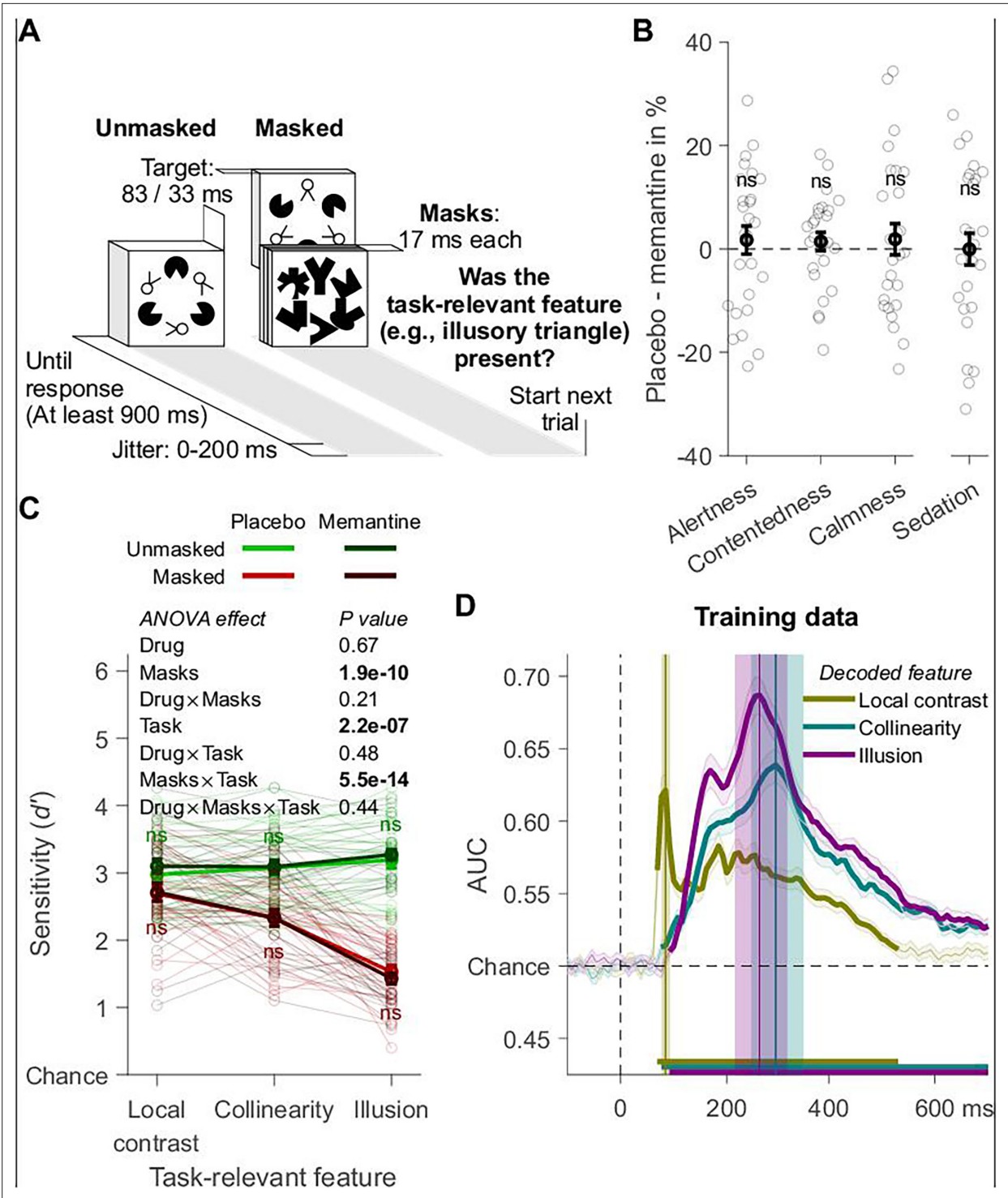

**Figure 4.** Experiment 2: design, behavior, and training data. (**A**) Trial design. (**B**) Effects of memantine on subjective state assessed with the visual analog scale (VAS; *n* = 26, one fewer than in the other experiment 2 analyses due to incomplete VAS data). The categories alertness, contentedness, and calmness are based on *Bond and Lader, 1974*, sedation on *Danion et al., 1989*. For details, see Methods. (**C**) Participants' ability to detect the task-relevant feature. (**D**) Tenfold cross-validation decoding scheme applied to an independent training dataset. Mean decoding performance, area under the receiver operating characteristic curve (AUC), over time ± standard error of the mean (SEM) is shown. Thick lines differ from chance: p < 0.05, cluster-based permutation test. Local contrast, collinearity, and illusion decoding accuracy peaked at 86, 297, and 266 ms after stimulus onset, and these peaks were encompassed by a 78–94, 247–347, and 217–317 ms time window, respectively. Error bars are mean ± SEM. Individual data points are plotted using low contrast. Ns is not significant (p > 0.05).

*Figure 4 continued on next page*

*Figure 4 continued*

The online version of this article includes the following figure supplement(s) for figure 4:

**Figure supplement 1.** Stimulus set of experiment 2.

**Figure supplement 2.** Bias.

**Figure supplement 3.** Behavior over the course of the experiments, divided into four time bins.

we employed less strong masking in this experiment than in experiment 1, leading to overall above-chance perceptual sensitivity for the masked trials. For unmasked trials, performance was at ceiling for all visual features (no differences in performance). Perceptual sensitivity was unaffected by memantine (*Figure 4C*).

The EEG decoding schemes were the same as for experiment 1, and similar time windows were obtained from the independent training dataset (*Figure 4D*). The localizer task was similar to the main experiment, but no masking was used. This localizer task was performed in all three sessions (intake and two main sessions involving the pharmacological manipulation, 1728 trials per session). In the intake session, no drugs were administered during the localizer task. In the other two sessions, the localizer task was performed right after pill intake, and this was long before memantine reached peak plasma levels (after ~4 hr). Overall, the timing of the decoding of the three different features in the main experiment replicated our findings from experiment 1 (*Figure 5*). For local contrast, task relevance had no effect, and the significant drop in decoding accuracy caused by masking was again numerically small (0.010 AUC on average); smaller than for collinearity (0.015 AUC) and illusion decoding (0.069 AUC). Collinearity (*Figure 5C, D*) and illusion (*Figure 5E, F*) decoding were impaired by masking and boosted when the corresponding feature was task-relevant. Crucially, our cluster-based analyses again revealed that memantine significantly improved decoding only for the illusion and only when the illusion was unmasked and task-relevant (*Figure 5E*, top left), thus replicating the results from experiment 1. Also, the time window-based analysis revealed an overall main effect of drug for illusion decoding, and post hoc tests revealed that the drug effect was only significant when the illusion was unmasked and task-relevant (post hoc *t*-tests comparing memantine vs. placebo, $t_{28}$ = −3.36, p = 0.002, $BF_{10}$ = 15.66; *Figure 5F*). For our time window-based analyses of illusion decoding, the specificity of the memantine effect to the unmasked condition was supported by a significant interaction between drug and masking (note, however, given overall much lower decoding accuracy in the masked condition, the lack of a memantine effect could reflect a floor effect). However, we did not obtain a statistically significant interaction between memantine and task relevance. Thus, although the memantine effect was significant only when the illusion was unmasked and task-relevant, just like for the effect of temporal attention in experiment 1, these results do not conclusively demonstrate a selective effect of memantine that depends on attention (task-relevance). Our time window-based ANOVAs now also revealed a main effect of memantine on collinearity decoding (*Figure 5D*), in line with the hypothesized role of NMDA receptors in recurrent (feedback and lateral) processing.

## Discussion

In this study, we investigated the impact of NMDA receptor blockade on multivariate EEG decoding of different features of a visual stimulus (Kanizsa illusion, collinearity, and local contrast). We used the NMDA receptor antagonist memantine to selectively target recurrent processing while minimizing non-specific side effects, a notable improvement over previous studies blocking NMDA receptors using ketamine. Two separate experiments demonstrated selective effects of memantine on decoding accuracy of the Kanizsa illusion, our marker of feedback processing, while leaving collinearity and contrast decoding relatively unaffected, our markers for lateral and feedforward processing. Interestingly, in both experiments, the effect of memantine on illusion decoding tended to be specific to attended, task-relevant stimuli presented without a backward mask. In experiment 1, the illusion was task-relevant, and the effect was only found for stimuli presented unmasked and outside the AB, when full attention was available. In experiment 2, we obtained the effect only when the stimulus was unmasked and the illusion was task-relevant. Notably, rather than impairing illusion decoding, in both experiments, memantine improved illusion decoding. These findings provide new insight into the

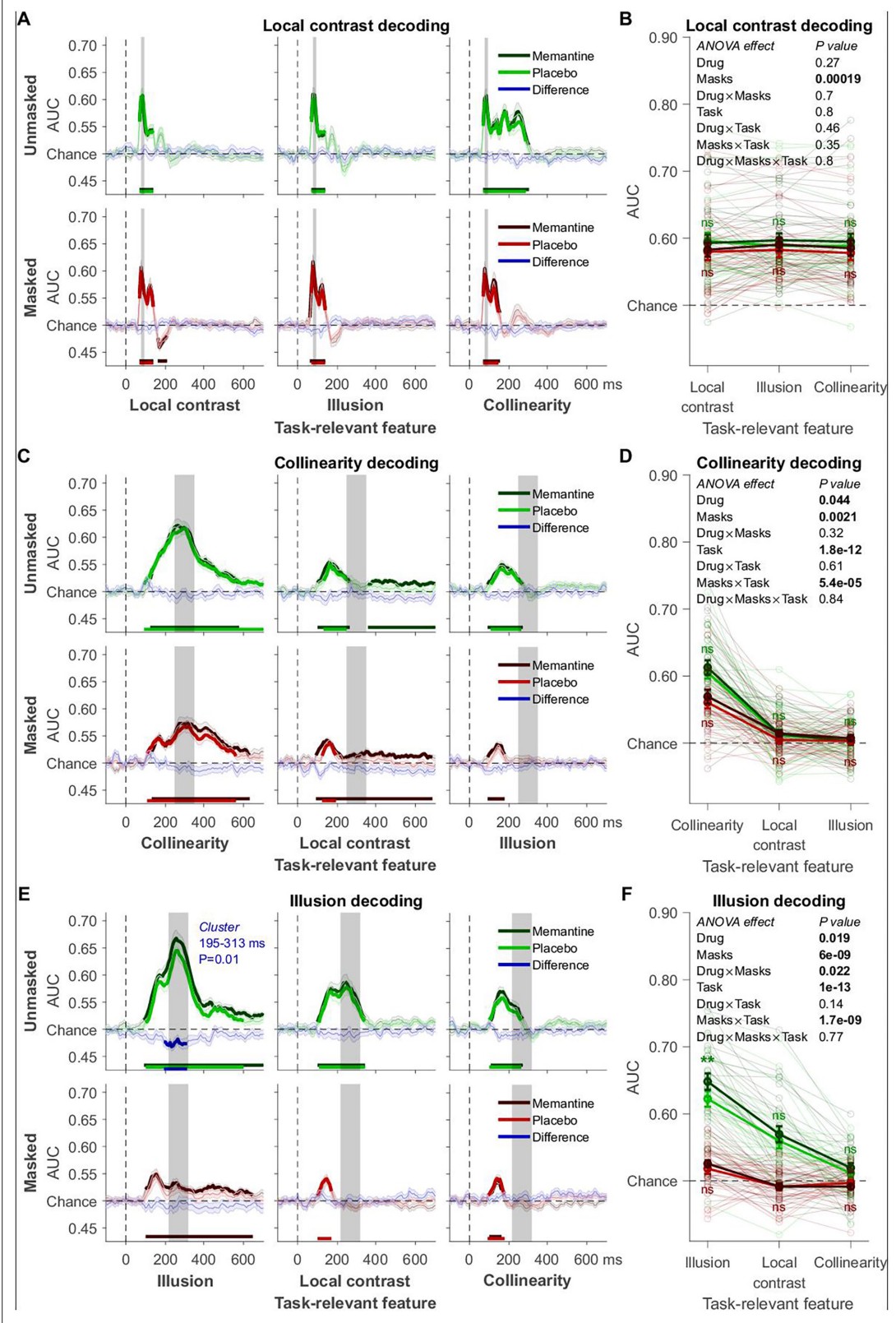

**Figure 5.** Decoding results of experiment 2. (**A**) Local contrast decoding: mean decoding performance, area under the receiver operating characteristic curve (AUC), over time ± standard error of the mean (SEM; n = 27). Thick lines differ from chance: p < 0.05, cluster-based permutation test. (**B**) Mean AUC for local contrast decoding's time window: 78–94 ms. Error bars are mean ± SEM. Individual data points are plotted using low contrast. Ns is not

*Figure 5 continued on next page*

*Figure 5 continued*

significant (p > 0.05). (**C**) Collinearity decoding: mean AUC over time. (**D**) Mean AUC for collinearity decoding's time window: 217–317 ms. (**E**) Kanizsa illusion decoding: mean AUC over time. (**F**) Mean AUC for Kanizsa illusion's time window: 223–323 ms. **p = 0.002.

The online version of this article includes the following figure supplement(s) for figure 5:

**Figure supplement 1.** Diagonal decoding results of experiment 2.

role of NMDA-receptor-mediated recurrent processing in human visual perception and will be further discussed below.

Our approach relied on using EEG decoding of different stimulus features at different points in time, together with their topography, as markers of distinct neural processes. Although such non-invasive, indirect measures of neural activity cannot provide direct evidence for feedforward vs. recurrent processes, the timing, topography, and susceptibility to masking of the decoding signatures obtained in the present study are consistent with neurophysiology (e.g., *Bosking et al., 1997*; *Kandel et al., 2000*; *Lamme and Roelfsema, 2000*; *Lee and Nguyen, 2001*; *Liang et al., 2017*; *Pak et al., 2020*), as well as with our previous work (*Fahrenfort et al., 2017*; *Noorman et al., 2023*). The specificity of the effect of memantine on our EEG marker of illusion-related feedback processing is broadly consistent with previous studies indicating that ketamine, another NMDA receptor antagonist, affects perception and neural coding in visual tasks thought to rely on recurrent mechanisms (*Meuwese et al., 2013*; *van Loon et al., 2016*). The effect of ketamine in these studies likely affected intra-area lateral connections in early visual cortex (*Meuwese et al., 2013*) and inter-area feedback mechanisms (*van Loon et al., 2016*), respectively. The present effect of memantine was largely specific to illusion decoding, our marker of feedback processing, while collinearity decoding, our marker of lateral processing, was not (experiment 1) or only weakly (experiment 2) affected by memantine. We have no straightforward explanation for why NMDA receptor blockade would impact inter-areal feedback connections more strongly than intra-areal lateral connections, considering their strong functional interdependency and interaction in grouping and segmentation processes (*Liang et al., 2017*). One possibility is that this finding reflects properties of our EEG decoding markers for feedback vs. lateral processing: for example, decoding of the Kanizsa illusion may have been more sensitive to the relatively subtle effect of our pharmacological manipulation, either because overall decoding was better than for collinearity or because NMDA receptor-dependent recurrent processes more strongly contribute to illusion decoding than to collinearity decoding.

While the effect of memantine in our study was largely specific to illusion decoding, the direction of this effect was in the opposite direction of what we had initially predicted and of the effect of ketamine in previous studies on perceptual representation: whereas ketamine impaired perceptual organization and visual cortex representations (*Meuwese et al., 2013*; *van Loon et al., 2016*), memantine improved illusion decoding. Although both drugs are known to inhibit NMDA receptors by occupying the receptor's ion channel and are thereby blocking current flow (*Glasgow et al., 2017*; *Molina et al., 2020*), the drugs have different actions at receptors other than NMDA, with ketamine acting on dopamine D2 and serotonin 5-HT2 receptors, and memantine inhibiting several subtypes of the acetylcholine (ACh) receptor as well as serotonin 5-HT3 receptors. Memantine and ketamine are also known to target different NMDA receptor subpopulations, with their inhibitory action displaying different time courses and intensity (*Glasgow et al., 2017*; *Johnson et al., 2015*). Blockade of different NMDA receptor subpopulations can result in markedly different and even opposite results. For example, *Self et al., 2012* found overall reduced or elevated visual activity after micro-infusion of two different selective NMDA receptor antagonists (2-amino-5-phosphonovalerate and ifendprodil) in macaque primary visual cortex. Although both drugs impaired the feedback-related response to figure vs. ground, similar to the effects of ketamine (*Meuwese et al., 2013*; *van Loon et al., 2016*), overall activity suggests that the effects of NMDA antagonism strongly depend on the targeted receptor subpopulation, each with distinct functional properties.

Indeed, previous studies have also found opposite effects of memantine and ketamine. For example, while ketamine diminishes the amplitude of the so-called mismatch negativity (MMN) EEG response to (auditory) oddball stimuli (*Rosburg and Kreitschmann-Andermahr, 2016*), a single dose of memantine enhances MMN amplitude (*Korostenskaja et al., 2007*; *Light et al., 2017*; *Swerdlow et al., 2016*). Although memantine was found to impair learning as well as markers of perceptual and neural processing in some tasks (e.g., *Rammsayer, 2001*; *Rammsayer, 2006*; *Schugens et al., 1997*), most

studies found memantine to improve learning and to boost perceptual and neural processing (*Golisch et al., 2017*; *Korostenskaja et al., 2007*; *Light et al., 2017*; *Swerdlow et al., 2016*; *Swerdlow et al., 2020*). As these previous studies typically used auditory stimuli, our results with visual stimuli add to a growing body of literature indicating that memantine can enhance sensory function, instead of decreasing it, and is the first study explicitly linking memantine to recurrent processing.

At this point, we can only speculate why NMDA receptor blockade via memantine increased (rather than decreased) decoding accuracy in the condition measuring illusion-specific feedback. One speculative possibility is that NMDA receptor blockade may have reduced signal variability or entropy, resulting in more stereotypical neural processing, which may have improved classifier performance. The effect of memantine on the excitation/inhibition (E/I) balance in human participants with schizophrenia, as measured noninvasively by computing the aperiodic, 1/f profile of the EEG power spectrum (*Molina et al., 2020*), may provide one explanation. A flatter 1/f power spectrum is thought to reflect noisier signal processing. Changes in E/I balance have been linked to variations in states of consciousness (*He et al., 2010*; *Nir et al., 2008*), and may also affect decoding performance due to changes in neural noise. As mentioned in the Introduction, this may be related to memantine modulating processing at other pre- or post-synaptic receptors present at NMDA-rich synapses, specifically affecting extrasynaptic NMDA receptors in GABAergic cells (*Milnerwood et al., 2010*; *Yao et al., 2022*). Memantine's strong effect on extrasynaptic NMDA receptors in GABAergic cells leads to increases in ACh levels, which have been shown to increase firing rates and reduce firing rate variability in macaques (*Herrero et al., 2013*; *Herrero et al., 2008*). This may represent a mechanism through which memantine (but not ketamine or the NMDA receptor antagonists used by Self and colleagues) could boost visually evoked activity. Clearly, further research is needed to elucidate the precise mechanisms by which memantine modulates feedback processing, how EEG decoding may relate to signal variability, and to explore its implications for perception.

However, our behavioral results show that improved decoding accuracy under memantine was not associated with boosted conscious perception. There are several possible reasons for this lack of behavioral correlate. For example, EEG decoding may be a more sensitive measure of the neural effects of memantine, in particular given that perceptual sensitivity may have been at floor (masked condition, experiment 1) or ceiling (unmasked condition, experiment 1, and experiment 2). It is also possible that the present decoding results are merely epiphenomenal, not mapping onto functional improvements (e.g., *Williams et al., 2007*). However, given that we found a tight link between these EEG decoding markers and behavioral performance in our previous work (*Fahrenfort et al., 2017*; *Noorman et al., 2023*), it is possible that the effect of memantine was just too subtle to show up in changes in overt behavior. Similarly, although we cannot fully exclude that the effect of memantine reflected differential perceptual learning via memantine's impact on LTP, we did not find evidence for such an effect based on perceptual learning: dividing the experiments in four time bins did not reveal differences or changes over time in illusion detection between placebo and memantine (see *Figure 4—figure supplement 3*).

Interestingly, we found an effect of memantine on illusion decoding only for temporally attended (experiment 1, AB task) and task-relevant Kanizsa stimuli (both experiments) that were presented without a backward mask. Although these results have to be interpreted with caution because the key interaction effects were not statistically significant, this suggests that object-based attention is crucial for memantine to exert its effects. Previous work has revealed that object-based attention primarily enhances neuronal firing rates compared to unattended objects, especially during the recurrent processing phase (*Roelfsema, 2023*; *Thiele and Bellgrove, 2018*). Attention-related neural enhancement can be observed in many regions throughout the cortex, for example, in early and intermediate visual regions, as well as inferotemporal, parietal, and even frontal cortices (reviewed in *Roelfsema, 2023*). It is hypothesized that when an object is attended, this facilitates the integration of its local elements and that it enables figure-ground segmentation to occur, crucial for the perception of coherent shapes and for the phenomenological properties of the Kanizsa illusion (*Narayanan et al., 2013*). The observation that the effect of inhibiting NMDA receptors by memantine was only found for attended objects may suggest that memantine is most effective in conditions where recurrent processing is strongest. Previous work has shown that memantine also affects cholinergic receptors (*Glasgow et al., 2017*; *Johnson et al., 2015*), known to be involved in attentional processes (*Herrero et al., 2013*; *Herrero et al., 2008*; *Thiele and Bellgrove, 2018*), which may partly explain why the

effect of memantine might be dependent on attentional allocation. This preliminary finding deserves future investigation to further understand its mechanistic origin.

In conclusion, our study demonstrates that memantine exerts largely selective effects on feedback processing during human perceptual inference, and these effects are contingent upon the allocation of attention. These findings contribute to our understanding of the neural mechanisms underpinning consciousness and highlight the importance of considering attentional factors in the study of recurrent processing. Further research is needed to elucidate the precise mechanisms by which memantine modulates feedback processing and to explore its implications for our understanding of perceptual inference and consciousness.

## Methods
### Experiment 1
#### Participants
Thirty-three right-handed men with normal, or corrected-to-normal, vision were recruited after they had passed extensive physical and mental screening (for exclusion criteria see Appendix 1). Three of them failed at the screening session's training task and did not proceed to the two experimental sessions (see Procedure). One dropped out during his first experimental session because of a headache, which was unrelated to memantine as he had taken the placebo. The remaining 29 participants (22 ± 3 years old) were analyzed. The study was approved by the Medical Ethical Committee of the Amsterdam Medical Centre (AMC) (NL64341.018.18) and the local ethics committee of the University of Amsterdam (2022-BC-14355). Participants gave informed consent and received monetary compensation.

#### Stimuli
The target stimulus set had a 2 Kanizsa illusion levels (absent/present) × 2 collinearity levels (absent/present) × 2 rotation levels (absent/present) design, resulting in eight stimuli (*Figure 1A*). Three aligned Pac-Men induced the Kanizsa illusion. Collinearity was present when the stimuli's three other elements (the 'two-legged white circles') were aligned. The controls for both the illusion and collinearity were made by rotating their elements 90°. Differences in local contrast were created by rotating the entire stimulus 180°. The targets spanned 7.5° by 8.3° of visual angle. The distance between the three Pac-Man stimuli as well as between the three aligned two-legged white circles was 2.8° of visual angle. Although neuronal responses to collinearity in primary visual cortex are most robust when this distance is smaller (*Kapadia et al., 1995*; *Kapadia et al., 2000*), longer-range horizontal connections between neurons with similar orientation selectivity can span distances corresponding to visual angles considerably greater than 2.8° (*Bosking et al., 1997*; *Stettler et al., 2002*).

The distractor stimulus set was the same as the target stimulus set, with two exceptions. First, the distractors were red instead of black. Second, the distractors' six elements were rotated 180° relative to the targets', so neither the illusion nor collinearity were present in the distractors. Masks consisted of six differently shaped elements, all capable of covering the targets' elements. Six masks were created by rotating the original mask five times by 60°. They spanned 8.5° by 9.1° of visual angle. Finally, the design of the fixation cross was adapted from *Thaler et al., 2013*.

#### Task designs
The training dataset for our decoding analyses was obtained from a separate 'template-matching' task performed in the intake session. No drugs were administered during this 'localizer' task. Here, every 900–1100 ms, a target stimulus was presented for 50 ms. Participants had to indicate whether the target matched the 'template', one of the eight target stimuli that was presented before the start of a block of trials. Each stimulus served as the template once. In short, the task design was 8 templates × (7 mismatches × 16 presentations + 1 match × 48 presentations), totaling 1280 trials, 3/10 of which were matches.

The main task was an RSVP task. Here, on every trial, participants were shown two targets (T1 and T2) within a RSVP of distractors (*Figure 1B*). The targets and distractors had a stimulus onset asynchrony of 100 ms. T2 and each distractor were presented for 17 ms. With the aim of inducing an AB, attending to T1 was made easier by presenting it for 67 ms. T1 was preceded by five distractors, and

T2 was followed by six distractors. For the first six participants, T2 and each distractor were presented for 10 ms and T1 for 40 ms (using the monitor's 100 Hz refresh rate). These durations were changed to make the task easier, so fewer volunteers would need to be excluded (see Procedure).

T2 visibility was manipulated in two ways. First, half the T2s were unmasked. The other half was immediately followed by three masks, each presented for 17 ms with an interstimulus interval of 0 ms. The three masks were selected randomly, but all differed from each other. Masking drastically reduced T2 visibility. Second, half the lags between the two targets were long: eight distractors (i.e., 900 ms). The other half was short: one or two distractors (i.e., 200 or 300 ms). The participant's short lag duration was determined beforehand, during the screening session (see Procedure). Short lags often induced an AB (i.e., missing T2s). Following the RSVP sequence, participants indicated for each target whether it contained the Kanizsa illusion or not. Response screens lasted until the response. Response button mapping was counterbalanced within the task. In short, each experimental session had an 8 T1 stimuli × 8 T2 stimuli × 2 masking levels (absent/present) × 2 lags (long/short) task design, resulting in 256 conditions. Each condition was presented four times, totaling 1024 trials.

For both the RSVP and the template-matching task, participants were instructed to fixate on the fixation cross.

## Procedure

The experiment consisted of three separate sessions conducted on different days: a 4-hr screening session and two 7-hr experimental sessions. At the beginning of the screening session, the volunteers were screened for contraindications to using memantine. If there were no contraindications, participants proceeded to performing the template-matching task with EEG and training for the RSVP task. To proceed to the experimental sessions, participants had to score well enough on this training task. Three participants were excluded for failing to achieve this. The training task was also used to discover the duration of the short lag that led to the lowest score to reliably induce an AB. For all participants, T2 performance at short lags was more than 5% lower than at long lags.

Memantine (20 mg) and the placebo were administered in different experimental sessions, separated by at least a week. These sessions were conducted using a randomized, double-blind crossover design. The order of drug administration (placebo or memantine first) was counterbalanced across participants. Participants were required to refrain from recreational drugs 4 weeks before the experiment and to refrain from alcohol 2 days before each experimental session. Experimental sessions started at 9:00 with the pill intake and briefly practicing the RSVP task. The actual RSVP task was performed when memantine reached peak plasma level, 4 hr after the pharmaceutical was administered. After this task, 5 min of resting state data were collected, and another task, the local–global task (*Bekinschtein et al., 2009*), was performed. These latter two paradigms are beyond the scope of this paper.

Before pill intake (~9:00) and before (~13:00) and after (~15:30) the tasks, participants filled in the VAS to measure the effects of memantine on subjective state (*Bond and Lader, 1974*; *Danion et al., 1989*). Tasks were programmed in Presentation software (Neurobehavioral Systems) and displayed on a 23-inch, 60 Hz, 1920 × 1080 pixels monitor, viewed at an 80 cm distance with the use of a chin rest.

## Behavioral analysis

To investigate behavioral performance on the RSVP task, we used perceptual sensitivity (*d'*) and bias (criterion, see *Figure 4—figure supplement 2*) derived from signal detection theory. The AB requires T1 to be attended, so trials with incorrect T1s were excluded when T2 performance was investigated. To investigate the VAS measurements, the mean scores of subsets of the scales were calculated and taken as measures of alertness, contentedness, and calmness following *Bond and Lader, 1974* and of sedation following *Danion et al., 1989*. VAS scores of the two measurements after pill intake were averaged and corrected by calculating percentage change from the baseline measurement. For all behavioral analyses, repeated-measures ANOVAs and double-sided (Bayesian) *t*-tests were used to test the differences between our conditions.

## EEG recording and preprocessing

EEG was recorded at 1024 Hz using a 64-channel ActiveTwo system (BioSemi). Four electrooculographic electrodes measured horizontal and vertical eye movements. The data were analyzed with

MATLAB (MathWorks). For most of the preprocessing steps, EEGLAB was used (*Delorme and Makeig, 2004*). The data were re-referenced to the earlobes. Poor channels were interpolated. High-pass filtering can cause artifacts in decoding analyses, so for slow-drift removal, we used trial-masked robust detrending (*van Driel et al., 2021*). Each trial was epoched from –250 to 1000 ms relative to target onset. To improve our independent component analysis (ICA), baseline correction was applied using the whole epoch as baseline (*Groppe et al., 2009*). ICA was used to remove blinks. Blink components were removed manually. Baseline correction was applied, now using a –250 to 0 ms window relative to target onset. Trials with values outside of a –300 to 300 μV range were removed. We used an adapted version of FieldTrip's ft_artifact_zvalue function to detect and remove trials with muscle artifacts (*Oostenveld et al., 2011*). As in the behavioral analyses, T2s preceded by incorrect T1s were excluded. Finally, the data were downsampled to 128 Hz.

## Multivariate pattern analysis

We used the Amsterdam Decoding and Modeling (ADAM) toolbox to perform multivariate pattern analyses on the EEG data (*Fahrenfort et al., 2018*). Using a tenfold cross-validation scheme, we first analyzed the template-matching task's data. We decoded each visual feature (e.g., the Kanizsa illusion) by training linear discriminant classifiers to discriminate between the feature's absence and presence based on the raw EEG activity (across all electrodes). Individual participants' data were split into ten equal-sized folds after randomizing the task's trial order. A classifier was then trained on nine folds and tested on the tenth one, ensuring independence of the training and testing sets. This procedure was repeated until each fold served as the test set once. Classifier performance, here AUC (*Hand and Till, 2001*), was averaged across all ten iterations. We applied 'diagonal decoding': classifier performance was calculated for every time sample, and the classifiers were tested on the same time sample they had been trained on.

Each decoded visual feature showed a clear peak in classifier performance over time (*Figure 1E*). Many of our analyses focus on the time windows encompassing these peaks: from 78 to 94 ms for local contrast, 192–292 ms for collinearity, and 223–323 ms for the illusion (following the same procedure to determine the different time windows as in our previous work, see *Noorman et al., 2023*). To obtain topographic maps showing the neural sources of the classifier performance, we multiplied the classifier weights with the data covariance matrix, yielding covariance/class separability maps (*Haufe et al., 2014*). The covariance/class separability maps of the three time windows showed that the classifiers depended on occipitoparietal electrodes during peak decoding accuracy (*Figure 2A*). We restricted the decoding analyses to using only these electrodes to ensure that any effects we observed were not due to poor signal-to-noise ratio.

Next, classifiers were trained on the template-matchings task's data and tested on each condition of the RSVP task's data. We averaged each visual feature's time window in the training set's raw EEG data to increase power. Classifiers were trained on these averages and tested on all time samples of the test set, resulting in 'off-diagonal decoding' (*King and Dehaene, 2014*). To establish the putative markers for the different forms of neural processing, each visual feature was decoded in this manner. When decoding a visual feature, the average of the same feature's time window was used as the training set. Off-diagonal decoding allowed us to focus on the neural processes of interest. For completeness, we also performed a diagonal decoding version of this train–test scheme. Here, classifiers trained on a time sample of the template-matching task's data were tested on the same time sample of the RSVP task's data. The diagonal decoding results were qualitatively similar to the off-diagonal ones (*Figure 2—figure supplement 1*, *Figure 3—figure supplement 1*, and *Figure 5—figure supplement 1*).

For the time samples from –100 to 700 ms relative to target onset, we used a double-sided *t*-test to evaluate whether classifier performance differed from chance and whether the memantine conditions' classifier performance differed from the placebo conditions'. We used cluster-based permutation testing (10,000 iterations at a threshold of 0.05) to correct for multiple comparisons (*Maris and Oostenveld, 2007*). As in the behavioral analyses, repeated-measures ANOVAs and double-sided (Bayesian) *t*-tests—here applied to the time windows' average classifier performance—were used to test the differences between our conditions.

## Experiment 2

### Participants

Thirty right-handed men with normal, or corrected-to-normal, vision were recruited after they had passed extensive physical and mental screening. Two of them dropped out after the first experimental session because of personal reasons unrelated to memantine. EEG recording failed during an experimental session of one participant. The remaining 27 participants (22 ± 2 years old) were analyzed. The study was approved by the Medical Ethical Committee of the Amsterdam Medical Centre (AMC) and the local ethics committee of the University of Amsterdam. Participants gave informed consent and received monetary compensation.

### Stimuli

Stimuli were the same as in experiment 1, except for a small change to the visual feature collinearity, for which we removed the 'feet' from the two-legged white circles (*Figure 4—figure supplement 1*).

### Task designs

The training dataset for our decoding analyses was obtained from a separate 'localizer' task, performed in all three sessions. In the intake, no drugs were administered. In the other sessions, the localizer was performed well before memantine reached peak plasma levels. Here, in separate blocks of trials, each visual feature was task-relevant. On every trial, a target stimulus was presented for 83 ms. Participants had to fixate on the fixation cross and indicate whether the current task-relevant feature (e.g., the Kanizsa illusion) was present or absent. The (first) response window started at target onset and ended after 900 ms. If no response was made, 'Respond faster!' was presented until response. Trials ended with 0–200 ms jitter and at least 300 ms between response and jitter. The order of the task-relevant features was counterbalanced over participants, and because each participant performed the task multiple times, it kept the same within participants. Response button mapping was counterbalanced within the task. In short, the task design was 3 task-relevant features × 8 target stimuli × 72 presentations, totaling 1728 trials. Participants completed the task three times in separate sessions, for a total of 5184 trials (1728 trials per session). All trials were used to build the classifiers.

The main task the participants performed was the 'task-relevance' task (*Figure 4A*), which was the same as the localizer task, except for the following changes. Half the trials were masked, and on these trials, the target stimulus was presented for 33 ms and immediately followed by three masks, the same way as in the RSVP task. As in the localizer task, each visual feature was at one point task-relevant and thus attended, while at another point task-irrelevant and thus less attended. The task also contained a base-rate manipulation, whose effects are beyond the scope of this paper. In one block of trials, the task-relevant feature was present on 75% of the trials, while on the next block, it was present on 25% of the trials, or vice versa. Overall, the task-relevant feature was equally often present and absent, which made it possible to ignore the base-rate manipulation for the purpose of the present analyses. In short, the task design was 3 task-relevant features × 8 target stimuli × 2 masking levels (absent/present) × 80 presentations, totaling 3840 trials.

To achieve masked performance of ~75% correct (making it more comparable to a typical AB performance), we staircased mask contrast using the weighted up-down method (*Kaernbach, 1991*). Staircasing was done when the Kanizsa illusion was task-relevant, because masking affected this feature more than the other two. Contrast levels ranged from 0 (black) to 255 (white). Mask contrast started at level 230. Each correct response made the task more difficult: masks got darker by downward step size $S_{down}$. Each incorrect response made the task easier: masks got lighter by upward step size $S_{up}$. Step sizes were determined by $S_{up} \times p = S_{down} \times (1 - p)$, where $p$ was the desired accuracy of 0.75. The smallest step size was always 13 contrast levels. The staircase ended after 25 reversals (a mistake after a correct response, or a correct response after a mistake). The mask contrast that was used during the experimental sessions was the average contrast level of the last 20 reversals.

### Procedure

The procedure was the same as for experiment 1, except for the following changes. During the screening session, if the volunteers had no contraindications to using memantine, they now proceeded to performing the localizer task and practicing the task-relevance task. This practice included our

staircase procedure. The mask strength obtained here was used during the experimental sessions. To increase the power of the decoding analyses, the pill intake at the start of the experimental sessions was now immediately followed by another session of the localizer task. The localizer task was followed by briefly practicing the task-relevance task. The actual task-relevance task was performed 4 hr after the pharmaceutical was administered. No additional tasks were performed. The experiment was conducted in two different types of laboratories. For 18 participants, tasks were displayed on a 23-inch, 1920 × 1080 pixels monitor, and for the other participants, on a 27-inch, 2560 × 1440 pixels monitor.

## Behavioral analysis
Behavioral analysis was the same as for experiment 1. One of the baseline VAS forms of one participant was not filled in, so he was excluded from the VAS analysis.

## EEG recording and preprocessing
EEG recording and preprocessing were the same as for experiment 1.

## Multivariate pattern analysis
Multivariate pattern analysis was the same as for experiment 1, except for the three instances of the localizer task now together serving the role of the template-matching task and the task-relevance task serving the role of the RSVP task. When decoding a visual feature (e.g., the Kanizsa illusion), not all of the localizer task's trials were used, only those during which the same visual feature was task-relevant. The tenfold cross-validation scheme was applied to the localizer task, and each decoded visual feature again showed a clear peak in classifier performance over time (*Figure 4D*). Here, the time windows encompassing these peaks were from 78 to 94 ms for local contrast, 247 to 347 ms for collinearity, and 217 to 317 ms for the illusion. Covariance/class separability maps again showed that the classifiers depended on occipitoparietal electrodes during peak decoding accuracy, so our decoding analyses were restricted to using only these electrodes.

---

# Additional information

## Competing interests
Simon van Gaal: Reviewing editor, *eLife*. The other authors declare that no competing interests exist.

## Funding

| Funder | Grant reference number | Author |
|---|---|---|
| HORIZON EUROPE European Research Council | 10.3030/715605 | Simon van Gaal |

The funders had no role in study design, data collection, and interpretation, or the decision to submit the work for publication.

## Author contributions
Samuel Noorman, Conceptualization, Data curation, Software, Formal analysis, Investigation, Visualization, Methodology, Writing – original draft, Project administration, Writing – review and editing; Timo Stein, Conceptualization, Supervision, Methodology, Writing – original draft, Writing – review and editing; Jasper Zantvoord, Investigation, Methodology; Johannes Fahrenfort, Conceptualization, Software, Supervision, Methodology, Writing – review and editing; Simon van Gaal, Conceptualization, Resources, Software, Supervision, Funding acquisition, Methodology, Writing – review and editing

## Author ORCIDs
Samuel Noorman ⬢ https://orcid.org/0000-0003-0953-1070
Timo Stein ⬢ https://orcid.org/0000-0002-8484-0933
Jasper Zantvoord ⬢ https://orcid.org/0000-0002-6475-902X
Johannes Fahrenfort ⬢ https://orcid.org/0000-0002-9025-3436

Simon van Gaal [ID] https://orcid.org/0000-0001-6628-4534

### Ethics

Informed consent and consent to publish was obtained from all participants. All procedures were approved by the Medical Ethical Committee of the Amsterdam Medical Centre (AMC), under project number NL64341.018.18, and the ethics board of the Psychology Department of the University of Amsterdam, under project number 2022-BC-14355.

Reviewer #1 (Public review): https://doi.org/10.7554/eLife.100530.3.sa1
Reviewer #2 (Public review): https://doi.org/10.7554/eLife.100530.3.sa2
Reviewer #3 (Public review): https://doi.org/10.7554/eLife.100530.3.sa3
Author response https://doi.org/10.7554/eLife.100530.3.sa4

---

# Additional files

### Supplementary files

MDAR checklist

### Data availability

All code and data related to this paper are available at https://doi.org/10.21942/uva.28684382.

The following dataset was generated:

| Author(s) | Year | Dataset title | Dataset URL | Database and Identifier |
|---|---|---|---|---|
| Noorman SG | 2025 | A causal role of the NMDA receptor in recurrent processing during perceptual integration | https://doi.org/10.21942/uva.28684382 | UvA/AUAS figshare, 10.21942/uva.28684382 |

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

## Appendix 1

Participant exclusion criteria were:

- Allergy to memantine or one of the inactive ingredients of these products;
- (History of) psychiatric treatment;
- First-degree relative with (history of) schizophrenia or major depression;
- (History of) clinically significant hepatic, cardiac, obstructive respiratory, renal, cerebrovascular, metabolic, or pulmonary disease, including, but not limited to fibrotic disorders;
- Claustrophobia;
- Regular usage of medicines (antihistamines or occasional use of paracetamol);
- (History of) neurological disease;
- (History of) epilepsy;
- Abnormal hearing or (uncorrected) vision;
- Average use of more than 15 alcoholic beverages weekly;
- Smoking;
- History of drug (opiate, LSD, (meth)amphetamine, cocaine, solvents, cannabis, or barbiturate) or alcohol dependence;
- Any known other serious health problem or mental/physical stress;
- Used psychotropic medication or recreational drugs over a period of 72 hr prior to each test session;
- Used alcohol within the last 24 hr prior to each test session;
- (History of) pheochromocytoma;
- Narrow-angle glaucoma;
- (History of) ulcer disease;
- Galactose intolerance, Lapp lactase deficiency, or glucose-galactose malabsorption.

