## [Editor Report · eLife Assessment]

This study presents a new, **fundamental** finding to the field interested in recurrent processing and its neuromodulatory underpinnings, finding unexpectedly that memantine (blocking NMDA-receptors) enhances the decoding of features thought to rely on NMDA-receptors. This interesting, **compelling** result identifies new directions for researchers studying consciousness, sensory processing, attention, and neurotransmitters.

---

## [Referee Report · Reviewer #1 (Public review)]

The authors investigate the function and neural circuitry of reentrant signals in visual cortex. Recurrent signaling is thought to be necessary to common types of perceptual experience that are defined by long-range relationships or prior expectation. Contour illusions - where perceptual objects are implied by stimuli characteristics - are a good example of this. The perception of these illusions is thought to emerge as recurrent signals from higher cortical areas feedback onto early visual cortex, to tell early visual cortex that it should be seeing object contours where none are actually present.

The authors test the involvement of reentrant cortical activity in this kind of perception using a drug challenge. Reentrance in visual cortex is thought to rely on NMDAR-mediated glutamate signalling. The authors accordingly employ an NMDA antagonist to stop this mechanism, looking for the effect of this manipulation on visually evoked activity recorded in EEG.

The motivating hypothesis for the paper is that NMDA antagonism should stop recurrent activity, and that this should degrade perceptual activity supporting perception of a contour illusion, but not other types of visual experience. Results in fact show the opposite. Rather than degrading cortical activity evoked by the illusion, memantine makes it more likely that machine learning classification of EEG will correctly infer the presence of the illusion.

On the face of it, this is confusing. But the paper does a good job of providing possible accounts based on specific details of neurochemical signalling and receptor populations.

I broadly find the paper interesting, graceful, and creative. The hypotheses are clear and compelling, the techniques for both manipulation of brain state and observation of that impact are cutting edge and well suited, and the paper draws clear and convincing conclusions that are made necessary by the results. The work sits at the very interesting crux of systems neuroscience, neuroimaging, and pharmacology.

---

## [Referee Report · Reviewer #2 (Public review)]

This study presents an important finding to the field interested in recurrent processing and the role of NMDA-receptors herein. The evidence for improved decoding under memantine is convincing, while some open questions remain to be followed up in future studies (the lack of a behavioural effect, why is decoding improved rather than decreased?). It is an excellent example of how an unexpected finding can generate novel research ideas to the mechanisms underlying recurrent processing, suggesting that the answer lies in the differences in the effects of ketamine and memantine, rather than their commonalities.

I would like to thank the authors for the great care they have taken in addressing my concerns. I think the revised manuscript is significantly easier to follow now that specific hypothesis have been formulated in the introduction, and the direction of the results is explicitly stated throughout the manuscript. I further appreciate the dampening of some of the claims that are not completely supported by the appropriate interactions.

I think the resulting manuscript is an incredibly exciting contribution to our understanding of NMDA-receptor function, and a great example of how an unexpected finding can raise questions that could potentially drive the field forward. It shows how NMDA's role in recurrent processing is much more complicate than previously assumed, and reveals that it is not the commonalities between memantine and ketamine that are important in understanding recurrent processing, but rather the differences. I look forward to future studies that will target these differences.

Overall great job.

---

## [Referee Report · Reviewer #3 (Public review)]

Summary:

In this study, Stein and colleagues use a clever masking/attentional blink paradigm using Kanisza stimuli, coupled with EEG decoding and the NMDA antagonist memantine, to isolate putative neural markers of feedforward, lateral, and feedback processing.

In two elegant experiments, they show that memantine selective influences EEG decoding of only illusory Kanisza surfaces (but not contour continuation or raw contrast), only when unmasked, only when attention is available (not when "blinked"), and only when task-relevant.

This neatly implicates NMDA receptors in the feedback mechanisms that are believed to be involved in inferring illusory Kanisza surfaces, and builds a difficult bridge between the large body of human perceptual experiments and pharmacological and neurophysiological work in animals.

Strengths:

Three key strengths of the paper are (1) its elegant and thorough experimental design, which includes internal replication of some key findings, and (2) the clear pattern of results across the full set of experiments, and (3) its clear writing and presentation of results.

The paper effectively reports a 4-way interaction, with memantine only influencing decoding of surfaces (1) that are unmasked (2), with attention available (3) and task-relevant (4). Nevertheless, the results are very clear, with a clear separation between null effects on other conditions and quite a strong (and thus highly selective) effect on this one intersection of conditions. This makes the pattern of findings very convincing.

Weaknesses:

Overall this is an impressive and important paper. However, to my mind there are two minor weaknesses.

First, despite its clear pattern of neural effects, there is no corresponding perceptual effect. Although the manipulation fits neatly within the conceptual framework, and there are many reasons for not finding such an effect (floor and ceiling effects, narrow perceptual tasks etc), this does leave open the possibility that the observation is entirely epiphenomenal, and that the mechanisms being recorded here are not actually causally involved in perception per se.

Second, although it is clear that there is an effect on decoding in this particular condition, what that means is not entirely clear - particularly since performance improves, rather than decreases. It should be noted here that improvements in decoding performance do not necessarily need to map onto functional improvements, and we should all be careful to remain agnostic about what is driving classifier performance. Here too, the effect of memantine on decoding might be epiphenomenal - unrelated to the information carried in the neural population, but somehow changing the balance of how that is electrically aggregated on the surface of the skull. *Something* is changing, but that might be a neurochemical or electrical side-effect unrelated to actual processing (particularly since no corresponding behavioural impact is observed.)

Comments on revisions:

I think the authors responsed fairly to my comments. Even if they weren't really able to add new insight into why behaviour didn't show the same effects as decoding, they discuss this in the revised text.

---

## [Author Response]

The following is the authors’ response to the original reviews

**Reviewer #1:**
(1) I miss some treatment of the lack of behavioural correlate. What does it mean that metamine benefits EEG classification accuracy without improving performance? One possibility here is that there is an improvement in response latency, rather than perceptual sensitivity. Is there any hint of that in the RT results? In some sort of combined measure of RT and accuracy?

First, we would like to thank the reviewer for their positive assessment of our work and for their extremely helpful and constructive comments that helped to significantly improve the quality of our manuscript.

The reviewer rightly points out that, to our surprise, we did not obtain a correlate of the effect of memantine in our behavioral data, neither in the reported accuracy data nor in the RT data. We do not report RT results as participants were instructed to respond as accurately as possible, without speed pressure. We added a paragraph in the discussion section to point to possible reasons for this surprising finding:

“There are several possible reasons for this lack of behavioral correlate. For example, EEG decoding may be a more sensitive measure of the neural effects of memantine, in particular given that perceptual sensitivity may have been at floor (masked condition, experiment 1) or ceiling (unmasked condition, experiment 1, and experiment 2). It is also possible that the present decoding results are merely epiphenomenal, not mapping onto functional improvements (e.g., Williams et al., 2007). However, given that we found a tight link between these EEG decoding markers and behavioral performance in our previous work (Fahrenfort et al., 2017; Noorman et al., 2023), it is possible that the effect of memantine was just too subtle to show up in changes in overt behavior.”

(2) An explanation is missing, about why memantine impacts the decoding of illusion but not collinearity. At a systems level, how would this work? How would NMDAR antagonism selectively impact long-range connectivity, but not lateral connectivity? Is this supported by our understanding of laminar connectivity and neurochemistry in the visual cortex?

We have no straightforward or mechanistic explanation for this finding. In the revised discussion, we are highlighting this finding more clearly, and included some speculative explanations:

“The present effect of memantine was largely specific to illusion decoding, our marker of feedback processing, while collinearity decoding, our marker of lateral processing, was not (experiment 1) or only weakly (experiment 2) affected by memantine. We have no straightforward explanation for why NMDA receptor blockade would impact inter-areal feedback connections more strongly than intra-areal lateral connections, considering their strong functional interdependency and interaction in grouping and segmentation processes (Liang et al., 2017). One possibility is that this finding reflects properties of our EEG decoding markers for feedback vs. lateral processing: for example, decoding of the Kanizsa illusion may have been more sensitive to the relatively subtle effect of our pharmacological manipulation, either because overall decoding was better than for collinearity or because NMDA receptor dependent recurrent processes more strongly contribute to illusion decoding than to collinearity decoding.”

(3) The motivating idea for the paper is that the NMDAR antagonist might disrupt the modulation of the AMPA-mediated glu signal. This is in line with the motivating logic for Self et al., 2012, where NMDAR and AMPAR efficacy in macacque V1 was manipulated via microinfusion. But this logic seems to conflict with a broader understanding of NMDA antagonism. NMDA antagonism appears to generally have the net effect of increasing glu (and ACh) in the cortex through a selective effect on inhibitory GABAergic cells (eg. Olney, Newcomer, & Farber, 1999). Memantine, in particular, has a specific impact on extrasynaptic NMDARs (that is in contrast to ketamine; Milnerwood et al, 2010, Neuron), and this type of receptor is prominent in GABA cells (eg. Yao et al., 2022, JoN). The effect of NMDA antagonists on GABAergic cells generally appears to be much stronger than the effect on glutamergic cells (at least in the hippocampus; eg. Grunze et al., 1996).This all means that it's reasonable to expect that memantine might have a benefit to visually evoked activity. This idea is raised in the GD of the paper, based on a separate literature from that I mentioned above. But all of this could be better spelled out earlier in the paper, so that the result observed in the paper can be interpreted by the reader in this broader context.To my mind, the challenging task is for the authors to explain why memantine causes an increase in EEG decoding, where microinfusion of an NMDA antagonist into V1 reduced the neural signal Self et al., 2012. This might be as simple as the change in drug... memantine's specific efficacy on extrasynaptic NMDA receptors might not be shared with whatever NMDA antagonist was used in Self et al. 2012. Ketamine and memantine are already known to differ in this way.

We addressed the reviewer’s comments in the following way. First, we bring up our (to us, surprising) result already at the end of the Introduction, pointing the reader to the explanation mentioned by the reviewer:

“We hypothesized that disrupting the reentrant glutamate signal via blocking NMDA receptors by memantine would impair illusion and possibly collinearity decoding, as putative markers of feedback and lateral processing, but would spare the decoding of local contrast differences, our marker of feedforward processing. To foreshadow our results, memantine indeed specifically affected illusion decoding, but enhancing rather than impairing it. In the Discussion, we offer explanations for this surprising finding, including the effect of memantine on extrasynaptic NMDA receptors in GABAergic cells, which may have resulted in boosted visual activity.”

Second, as outlined in the response to the first point by Reviewer #2, we are now clear throughout the title, abstract, and paper that memantine “improved” rather than “modulated” illusion decoding.

Third, and most importantly, we restructured and expanded the Discussion section to include the reviewer’s proposed mechanisms and explanations for the effect. We would like to thank the reviewer for pointing us to this literature. We also discuss the results of Self et al. (2012), specifically the distinct effects of the two NMDAR antagonists used in this study, more extensively, and speculate that their effects may have been similar to ketamine and thus possibly opposite of memantine (for the feedback signal):

“Although both drugs are known to inhibit NMDA receptors by occupying the receptor’s ion channel and are thereby blocking current flow (Glasgow et al., 2017; Molina et al., 2020), the drugs have different actions at receptors other than NMDA, with ketamine acting on dopamine D2 and serotonin 5-HT2 receptors, and memantine inhibiting several subtypes of the acetylcholine (ACh) receptor as well as serotonin 5HT3 receptors. Memantine and ketamine are also known to target different NMDA receptor subpopulations, with their inhibitory action displaying different time courses and intensity (Glasgow et al., 2017; Johnson et al., 2015). Blockade of different NMDA receptor subpopulations can result in markedly different and even opposite results. For example, Self and colleagues (2012) found overall reduced or elevated visual activity after microinfusion of two different selective NMDA receptor antagonists (2-amino-5phosphonovalerate and ifendprodil) in macaque primary visual cortex. Although both drugs impaired the feedback-related response to figure vs. ground, similar to the effects of ketamine (Meuwese et al., 2013; van Loon et al., 2016) such opposite effects on overall activity demonstrate that the effects of NMDA antagonism strongly depend on the targeted receptor subpopulation, each with distinct functional properties.”

Finally, we link these differences to the potential mechanism via GABAergic neurons:

“As mentioned in the Introduction, this may be related to memantine modulating processing at other pre- or post-synaptic receptors present at NMDA-rich synapses, specifically affecting extrasynaptic NMDA receptors in GABAergic cells (Milnerwood et al, 2010; Yao et al., 2022). Memantine’s strong effect on extrasynaptic NMDA receptors in GABAergic cells leads to increases in ACh levels, which have been shown to increase firing rates and reduce firing rate variability in macaques (Herrero et al., 2013, 2008). This may represent a mechanism through which memantine (but not ketamine or the NMDA receptor antagonists used by Self and colleagues) could boost visually evoked activity.”

(4) The paper's proposal is that the effect of memantine is mediated by an impact on the efficacy of reentrant signaling in visual cortex. But perhaps the best-known impact of NMDAR manipulation is on LTP, in the hippocampus particularly but also broadly.Perception and identification of the kanisza illusion may be sensitive to learning (eg. Maertens & Pollmann, 2005; Gellatly, 1982; Rubin, Nakayama, Shapley, 1997); what argues against an account of the results from an effect on perceptual learning? Generally, the paper proposes a very specific mechanism through which the drug influences perception. This is motivated by results from Self et al 2012 where an NMDA antagonist was infused into V1. But oral memantine will, of course, have a whole-brain effect, and some of these effects are well characterized and - on the surface - appear as potential sources of change in illusion perception. The paper needs some treatment of the known ancillary effects of diffuse NMDAR antagonism to convince the reader that the account provided is better than the other possibilities.

We cannot fully exclude an effect based on perceptual learning but consider this possibility highly unlikely for several reasons. First, subjects have performed more than a thousand trials in a localizer session before starting the main task (in experiment 2 even more than two thousand) containing the drug manipulation. Therefore, a large part of putative perceptual learning would have already occurred before starting the main experiment. Second, the main experiment was counterbalanced across drug sessions, so half of the participants first performed the memantine session and then the placebo session, and the other half of the subjects the other way around. If memantine would have improved perceptual learning in our experiments, one may actually expect to observe improved decoding in the placebo session and not in the memantine session. If memantine would have facilitated perceptual learning during the memantine session, the effect of that facilitated perceptual learning would have been most visible in the placebo session following the memantine session. Because we observed improved decoding in the memantine session itself, perceptual learning is likely not the main explanation for these findings. Third, perceptual learning is known to occur for several stimulus dimensions (e.g., orientation, spatial frequency or contrast). If these findings would have been driven by perceptual learning one would have expected to see perceptual learning for all three features, whereas the memantine effects were specific to illusion decoding. Especially in experiment 2, all features were equally often task relevant and in such a situation one would’ve expected to observe perceptual learning effects on those other features as well.

To further investigate any potential role of perceptual learning, we analyzed participants’ performance in detecting the Kanizsa illusion over the course of the experiments. To investigate this, we divided the experiments’ trials into four time bins, from the beginning until the end of the experiment. For the first experiment’s first target (T1), there was no interaction between the factors bin and drug (memantine/placebo; *F*_3,84_=0.89, *P*=0.437; Figure S6A). For the second target (T2), we performed a repeatedmeasures ANOVA with the factors bin, drug, T1-T2 lag (short/long), and masks (present/absent). There was only a trend towards a bin by drug interaction (*F*_3,84_=2.57, *P*=0.064; Figure S6B), reflecting worse performance under memantine in the first three bins and slightly better performance in the fourth bin. The other interactions that include the factors bin and drug factors were not significant (all *P*>0.117). For the second experiment, we performed a repeated-measures ANOVA with the factors bin, drug, masks, and task-relevant feature (local contrast/collinearity/illusion). None of the interactions that included the bin and drug factors were significant (all *P*>0.219; Figure S6C). Taken together, memantine does not appear to affect Kanizsa illusion detection performance through perceptual learning. Finally, there was no interaction between the factors bin and task-relevant feature (*F*_6,150_=0.76, *P*=0.547; Figure S6D), implying there is no perceptual learning effect specific to Kanizsa illusion detection. We included these analyses in our revised Supplement as Fig. S6.

(5) The cross-decoding approach to data analysis concerns me a little. The approach adopted here is to train models on a localizer task, in this case, a task where participants matched a kanisza figure to a target template (E1) or discriminated one of the three relevant stimuli features (E2). The resulting model was subsequently employed to classify the stimuli seen during separate tasks - an AB task in E1, and a feature discrimination task in E2. This scheme makes the localizer task very important. If models built from this task have any bias, this will taint classifier accuracy in the analysis of experimental data. My concern is that the emergence of the kanisza illusion in the localizer task was probably quite salient, respective to changes in stimuli rotation or collinearity. If the model was better at detecting the illusion to begin with, the data pattern - where drug manipulation impacts classification in this condition but not other conditions - may simply reflect model insensitivity to non-illusion features.I am also vaguely worried by manipulations implemented in the main task that do not emerge in the localizer - the use of RSVP in E1 and manipulation of the base rate and staircasing in E2. This all starts to introduce the possibility that localizer and experimental data just don't correspond, that this generates low classification accuracy in the experimental results and ineffective classification in some conditions (ie. when stimuli are masked; would collinearity decoding in the unmasked condition potentially differ if classification accuracy were not at a floor? See Figure 3c upper, Figure 5c lower).What is the motivation for the use of localizer validation at all? The same hypotheses can be tested using within-experiment cross-validation, rather than validation from a model built on localizer data. The argument may be that this kind of modelling will necessarily employ a smaller dataset, but, while true, this effect can be minimized at the expense of computational cost - many-fold cross-validation will mean that the vast majority of data contributes to model building in each instance.It would be compelling if results were to reproduce when classification was validated in this kind of way. This kind of analysis would fit very well into the supplementary material.

We thank the reviewer for this excellent question. We used separate localizers for several reasons, exactly to circumvent the kind of biases in decoding that the reviewer alludes to. Below we have detailed our rationale, first focusing on our general rationale and then focusing on the decisions we made in designing the specific experiments.

Using a localizer task in the design of decoding analysis offers several key advantages over relying solely on k-fold cross-validation within the main task:

(1) Feature selection independence and better generalization: A separate localizer task allows for independent feature selection, ensuring that the features used for decoding are chosen without bias from the main task data. Specifically, the use of a localizer task allows us to determine the time-windows of interest independently based on the peaks of the decoding in the localizer. This allows for a better direct comparison between the memantine and placebo conditions because we can isolate the relevant time windows outside a drug manipulation. Further, training a classifier on a localizer task and testing it on a separate experimental task assesses whether neural representations generalize across contexts, rather than simply distinguishing conditions within a single dataset. This supports claims about the robustness of the decoded information.

(2) Increased sensitivity and interpretability: The localizer task can be designed specifically to elicit strong, reliable responses in the relevant neural patterns. This can improve signal-to-noise ratio and make it easier to interpret the features being used for decoding in the test set. We facilitate this by having many more trials in the localizer tasks (1280 in E1 and 5184 in E2) than in the separate conditions of the main task, in which we would have to do k-folding (e.g., 2, mask, x 2 (lag) design in E1 leaves fewer than 256 trials, due to preprocessing, for specific comparisons) on very low trial numbers. The same holds for experiment 2 which has a 2x3 design, but also included the base-rate manipulation. Finally, we further facilitate sensitivity of the model by having the stimuli presented at full contrast without any manipulations of attention or masking during the localizer, which allows us to extract the feature specific EEG signals in the most optimal way.

(3) Decoupling task-specific confounds: If decoding is performed within the main task using k-folding, there is a risk that task-related confounds (e.g., motor responses, attention shifts, drug) influence decoding performance. A localizer task allows us to separate the neural representation of interest from these taskrelated confounds.

Experiment 1

In experiment 1, the Kanizsa was always task relevant in the main experiment in which we employed the pharmacological manipulation. To make sure that the classifiers were not biased towards Kanizsa figures from the start (which would be the case if we would have done k-folding in the main task), we used a training set in which all features were equally relevant for task performance. As can be seen in figure 1E, which plots the decoding accuracies of the localizer task, illusion decoding as well as rotation decoding were equally strong, whereas collinearity decoding was weaker. It may be that the Kanizsa illusion was quite salient in the localizer task, which we can’t know at present, but it was at least less salient and relevant than in the main task (where it was the only task-relevant feature). Based on the localizer decoding results one could argue that the rotation dimension and illusion dimension were most salient, because the decoding was highest for these dimensions. Clearly the model was not insensitive to nonillusory features. The localizer task of experiment 2 reveals that collinearity decoding tends to be generally lower, even when that feature is task relevant.

Experiment 2

In experiment 2, the localizer task and main task were also similar, with three exceptions: during the localizer task no drug was active, and no masking and no base rate manipulation were employed. To make sure that the classifier was not biased towards a certain stimulus category (due to the bias manipulation), e.g. the stimulus that is presented most often, we used a localizer task without this manipulation. As can be seen in figure 4D decoding of all the features was highly robust, also for example for the collinearity condition. Therefore the low decoding that we observe in the main experiment cannot be due to poor classifier training or feature extraction in the localizer. We believe this is actually an advantage instead of a disadvantage of the current decoding protocol.

Based on the rationale presented above we are uncomfortable performing the suggested analyses using a k-folding approach in the main task, because according to our standards the trial numbers are too low and the risk that these results are somehow influenced by task specific confounds cannot be ruled out.

Line 301 - 'Interestingly, in both experiments the effect of memantine... was specific to... stimuli presented without a backward mask.' This rubs a bit, given that the mask broadly disrupted classification. The absence of memantine results in masked results may simply be a product of the floor ... some care is needed in the interpretation of this pattern.

In the results section of experiment 1, we added:

“While the interaction between masking and memantine only approached significance (*P*=0.068), the absence of an effect of memantine in the masked condition could reflect a floor effect, given that illusion decoding in the masked condition was not significantly better than chance.”

While floor is less likely to account for the absence of an effect in the masked condition in experiment 2, where illusion decoding in the masked condition was significantly above chance, it is still possible that to obtain an effect of memantine, decoding accuracy needed to be higher. We therefore also added here:

“For our time window-based analyses of illusion decoding, the specificity of the memantine effect to the unmasked condition was supported by a significant interaction between drug and masking (note, however, given overall much lower decoding accuracy in the masked condition, the lack of a memantine effect could reflect a floor effect).”

In the discussion, we changed the sentence to read “…the effect of memantine on illusion decoding *tended* to be specific to attended, task-relevant stimuli presented without a backward mask.”

Line 441 - What were the contraindications/exclusion parameters for the administration of memantine?

Thanks for spotting this. We have added the relevant exclusion criteria in the revised version of the supplement. See also below.

– Allergy for memantine or one of the inactive ingredients of these products;

– (History of) psychiatric treatment;

– First-degree relative with (history of) schizophrenia or major depression;

– (History of) clinically significant hepatic, cardiac, obstructive respiratory, renal, cerebrovascular, metabolic or pulmonary disease, including, but not limited to fibrotic disorders;

– Claustrophobia;

– Regular usage of medicines (antihistamines or occasional use of paracetamol);

– (History of) neurological disease;

– (History of) epilepsy;

– Abnormal hearing or (uncorrected) vision;

– Average use of more than 15 alcoholic beverages weekly;

– Smoking

– History of drug (opiate, LSD, (meth)amphetamine, cocaine, solvents, cannabis, or barbiturate) or alcohol dependence;

– Any known other serious health problem or mental/physical stress;

– Used psychotropic medication, or recreational drugs over a period of 72 hours prior to each test session,

– Used alcohol within the last 24 hours prior to each test session;

– (History of) pheochromocytoma.

– Narrow-angle glaucoma;

– (History of) ulcer disease;

– Galactose intolerance, Lapp lactase deficiency or glucose­galactose malabsorption.

– (History of) convulsion;

Line 587 - The localizer task used to train the classifier in E2 was collected in different sessions. Was the number of trials from separate sessions ultimately equal? The issue here is that the localizer might pick up on subtle differences in electrode placement. If the test session happens to have electrode placement that is similar to the electrode placement that existed for a majority of one condition of the localizer... this will create bias. This is likely to be minor, but machine classifiers really love this kind of minor confound.

Indeed, the trial counts in the separate sessions for the localizer in E2 were equal. We have added that information to the methods section.

Experiment 1: 1280 trials collected during the intake session.

In experiment 2: 1728 trials were collected per session (intake, and 2 drug sessions), so there were 5184 trials across three sessions.

**Reviewer #2:**
To start off, I think the reader is being a bit tricked when reading the paper. Perhaps my priors are too strong, but I assumed, just like the authors, that NMDA-receptors would disrupt recurrent processing, in line with previous work. However, due to the continuous use of the ambiguous word 'affected' rather than the more clear increased or perturbed recurrent processing, the reader is left guessing what is actually found. That's until they read the results and discussion finding that decoding is actually improved. This seems like a really big deal, and I strongly urge the authors to reword their title, abstract, and introduction to make clear they hypothesized a disruption in decoding in the illusion condition, but found the opposite, namely an increase in decoding. I want to encourage the authors that this is still a fascinating finding.

We thank the reviewer for the positive assessment of our manuscript, and for many helpful comments and suggestions.

We changed the title, abstract, and introduction in accordance with the reviewer’s comment, highlighting that “memantine […] improves decoding” and “enhances recurrent processing” in all three sections. We also changed the heading of the corresponding results section to “Memantine selectively improves decoding of the Kanizsa illusion”.

Apologies if I have missed it, but it is not clear to me whether participants were given the drug or placebo during the localiser task. If they are given the drug this makes me question the logic of their analysis approach. How can one study the presence of a process, if their very means of detecting that process (the localiser) was disrupted in the first place? If participants were not given a drug during the localiser task, please make that clear. I'll proceed with the rest of my comments assuming the latter is the case. But if the former, please note that I am not sure how to interpret their findings in this paper.

Thanks for asking this, this was indeed unclear. In experiment 1 the localizer was performed in the intake session in which no drugs were administered. In the second experiment the localizer was performed in all three sessions with equal trial numbers. In the intake session no drugs were administrated. In the other two sessions the localizer was performed directly after pill intake and therefore the memantine was not (or barely) active yet. We started the main task four hours after pill intake because that is the approximate peak time of memantine. Note that all three localizer tasks were averaged before using them as training set. We have clarified this in the revised manuscript.

The main purpose of the paper is to study recurrent processing. The extent to which this study achieves this aim is completely dependent to what extent we can interpret decoding of illusory contours as uniquely capturing recurrent processing. While I am sure illusory contours rely on recurrent processing, it does not follow that decoding of illusory contours capture recurrent processing alone. Indeed, if the drug selectively manipulates recurrent processing, it's not obvious to me why the authors find the interaction with masking in experiment 2. Recurrent processing seems to still be happening in the masked condition, but is not affected by the NMDA-receptor here, so where does that leave us in interpreting the role of NMDA-receptors in recurrent processing? If the authors can not strengthen the claim that the effects are completely driven by affecting recurrent processing, I suggest that the paper will shift its focus to making claims about the encoding of illusory contours, rather than making primary claims about recurrent processing.

We indeed used illusion decoding as a marker of recurrent processing. Clearly, such a marker based on a non-invasive and indirect method to record neural activity is not perfect. To directly and selectively manipulate recurrent processing, invasive methods and direct neural recordings would be required. However, as explained in the revised Introduction,

“In recent work we have validated that the decoding profiles of these features of different complexities at different points in time, in combination with the associated topography, can indeed serve as EEG markers of feedforward, lateral and recurrent processes (Fahrenfort et al., 2017; Noorman et al., 2023).”

The timing and topography of the decoding results of the present study were consistent with our previous EEG decoding studies (Fahrenfort et al., 2017; Noorman et al., 2023). This validates the use of these EEG decoding signatures as (imperfect) markers of distinct neural processes, and we continue to use them as such. However, we expanded the discussion section to alert the reader to the indirect and imperfect nature of these EEG decoding signatures as markers of distinct neural processes: “Our approach relied on using EEG decoding of different stimulus features at different points in time, together with their topography, as markers of distinct neural processes. Although such non-invasive, indirect measures of neural activity cannot provide direct evidence for feedforward vs. recurrent processes, the timing, topography, and susceptibility to masking of the decoding signatures obtained in the present study are consistent with neurophysiology (e.g., Bosking et al., 1997; Kandel et al., 2000; Lamme & Roelfsema, 2000; Lee & Nguyen, 2001; Liang et al., 2017; Pak et al., 2020), as well as with our previous work (Fahrenfort et al., 2017; Noorman et al., 2023).”

The reviewer is also concerned about the lack of effect of memantine on illusion decoding in the masked condition in experiment 2. In our view, the strong effect of masking on illusion decoding (both in absolute terms, as well as when compared to its effect on local contrast decoding), provides strong support for our assumption that illusion decoding represents a marker of recurrent processing. Nevertheless, as the reviewer points out, weak but statistically significant illusion decoding was still possible in the masked condition, at least when the illusion was task-relevant. As the reviewer notes, this may reflect residual recurrent processing during masking, a conclusion consistent with the relatively high behavioral performance despite masking (d’ > 1). However, rather than invalidating the use of our EEG markers or challenging the role of NMDA-receptors in recurrent processing, this may simply reflect a floor effect. As outlined in our response to reviewer #1 (who was concerned about floor effects), in the results section of experiment 1, we added:

“While the interaction between masking and memantine only approached significance (*P*=0.068), the absence of an effect of memantine in the masked condition could reflect a floor effect, given that illusion decoding in the masked condition was not significantly better than chance.”

And for experiment 1:

“For our time window-based analyses of illusion decoding, the specificity of the memantine effect to the unmasked condition was supported by a significant interaction between drug and masking (note, however, given overall much lower decoding accuracy in the masked condition, the lack of a memantine effect could reflect a floor effect).”

An additional claim is being made with regards to the effects of the drug manipulation. The authors state that this effect is only present when the stimulus is (1) consciously accessed, and (2) attended. The evidence for claim 1 is not supported by experiment 1, as the masking manipulation did not interact in the cluster-analyses, and the analyses focussing on the peak of the timing window do not show a significant effect either. There is evidence for this claim coming from experiment 2 as masking interacts with the drug condition. Evidence for the second claim (about task relevance) is not presented, as there is no interaction with the task condition. A classical error seems to be made here, where interactions are not properly tested. Instead, the presence of a significant effect in one condition but not the other is taken as sufficient evidence for an interaction, which is not appropriate. I therefore urge the authors to dampen the claim about the importance of attending to the decoded features. Alternatively, I suggest the authors run their interactions of interest on the time-courses and conduct the appropriate clusterbased analyses.

We thank the reviewer for pointing out the importance of key interaction effects. Following the reviewer’s suggestion, we dampened our claims about the role of attention. For experiment 1, we changed the heading of the relevant results section from “Memantine’s effect on illusion decoding requires attention” to “The role of consciousness and attention in memantine’s effect on illusion decoding”, and we added the following in the results section:

“Also our time window-based analyses showed a significant effect of memantine only when the illusion was both unmasked and presented outside the AB (_t_28=-2.76, *P*=0.010, BF_10_=4.53; Fig. 3F). Note, however, that although these post-hoc tests of the effect of memantine on illusion decoding were significant, for our time window-based analyses we did not obtain a statistically significant interaction between the AB and memantine, and the interaction between masking and memantine only approached significance (*P* = 0.068). Thus, although these memantine effects were slightly less robust than for T1, probably due to reduced trial counts, these results point to (but do not conclusively demonstrate) a selective effect of memantine on illusion-related feedback processing that depends on the availability of attention. In addition to the lack of the interaction effect, another potential concern…”

For experiment 2, we added the following in the results section:

“Note that, for our time window-based analyses of illusion decoding, although the specificity of the memantine effect to the unmasked condition was supported by a significant interaction between drug and masking, we did not obtain a statistically significant interaction between memantine and task-relevance. Thus, although the memantine effect was significant only when the illusion was unmasked and taskrelevant, just like for the effect of temporal attention in experiment 1, these results do not conclusively demonstrate a selective effect of memantine that depends attention (task-relevance).”

In the discussion, we toned down claims about memantine’s effects being specific to attended conditions, we are highlighting the “preliminary” nature of these findings, and we are now alerting the reader explicitly to be careful with interpreting these effects, e.g.:

“Although these results have to be interpreted with caution because the key interaction effects were not statistically significant, …”

How were the length of the peak-timing windows established in Figure 1E? My understanding is that this forms the training-time window for the further decoding analyses, so it is important to justify why they have different lengths, and how they are determined. The same goes for the peak AUC time windows for the interaction analyses. A number of claims in the paper rely on the interactions found in these posthoc analyses, so the 223- to 323 time window needs justification.

Thanks for this question. The length of these peak-timing windows is different because the decoding of rotation is temporarily very precise and short-lived, whereas the decoding of the other features last much longer and is more temporally variable. In fact, we have followed the same procedure as in a previously published study (Noorman et al., elife 2025) for defining the peak-timing and length of the windows. We followed the same procedure for both experiments reported in this paper, replicating the crucial findings and therefore excluding the possibility that these findings are in any way dependent on the time windows that are selected. We have added that information to the revised version of the manuscript.

**Reviewer #3:**
First, despite its clear pattern of neural effects, there is no corresponding perceptual effect. Although the manipulation fits neatly within the conceptual framework, and there are many reasons for not finding such an effect (floor and ceiling effects, narrow perceptual tasks, etc), this does leave open the possibility that the observation is entirely epiphenomenal, and that the mechanisms being recorded here are not actually causally involved in perception per se.

We thank the reviewer for the positive assessment of our work. The reviewer rightly points out that, to our surprise, we did not obtain a correlate of the effect of memantine in our behavioral data. We agree with the possible reasons for the absence of such an effect highlighted by the reviewer, and expanded our discussion section accordingly:

“There are several possible reasons for this lack of behavioral correlate. For example, EEG decoding may be a more sensitive measure of the neural effects of memantine, in particular given that perceptual sensitivity may have been at floor (masked condition, experiment 1) or ceiling (unmasked condition, experiment 1, and experiment 2). It is also possible that the present decoding results are merely epiphenomenal, not mapping onto functional improvements (e.g., Williams et al., 2007). However, given that in our previous work we found a tight link between these EEG decoding markers and behavioral performance (Fahrenfort et al., 2017; Noorman et al., 2023), it is possible that the effect of memantine in the present study was just too subtle to show up in changes in overt behavior.”

Second, although it is clear that there is an effect on decoding in this particular condition, what that means is not entirely clear - particularly since performance improves, rather than decreases. It should be noted here that improvements in decoding performance do not necessarily need to map onto functional improvements, and we should all be careful to remain agnostic about what is driving classifier performance. Here too, the effect of memantine on decoding might be epiphenomenal - unrelated to the information carried in the neural population, but somehow changing the balance of how that is electrically aggregated on the surface of the skull. *Something* is changing, but that might be a neurochemical or electrical side-effect unrelated to actual processing (particularly since no corresponding behavioural impact is observed.)

We would like to refer to our reply to the previous point, and we would like to add that in our previous work (Fahrenfort et al., 2017; Noorman et al., 2023) similar EEG decoding markers were often tightly linked to changes in behavioral performance. This indicates that these particular EEG decoding markers do not simply reflect some sideeffect not related to neural processing. However, as stated in the revised discussion section, “it is possible that the effect of memantine in the present study was just too subtle to show up in changes in overt behavior.”